# Noncanonical usage of stop codons in ciliates expands proteins with structurally flexible Q-rich motifs

**Chi-Ning Chuang[1†], Hou-Cheng Liu[1†], Tai-Ting Woo[1‡], Ju-Lan Chao[1],
Chiung-Ya Chen[1], Hisao-Tang Hu[1], Yi-Ping Hsueh[1,2], Ting-Fang Wang[1,2]***

[1]Institute of Molecular Biology, Academia Sinica, Taipei, Taiwan; [2]Department of
Biochemical Science and Technology, National Chiayi University, Chiayi, Taiwan

**\*For correspondence:**
tfwang@gate.sinica.edu.tw

[†]These authors contributed
equally to this work

**Present address:** [‡]Department of
Cell and Developmental Biology,
University of Michigan Medical
School, Ann Arbor, Michigan,
United States

**Competing interest:** The authors
declare that no competing
interests exist.

**Reviewing Editor:** Dominique
Soldati-Favre, University of
Geneva, Switzerland

**Abstract** Serine(S)/threonine(T)-glutamine(Q) cluster domains (SCDs), polyglutamine (polyQ)
tracts and polyglutamine/asparagine (polyQ/N) tracts are Q-rich motifs found in many proteins.
SCDs often are intrinsically disordered regions that mediate protein phosphorylation and protein-
protein interactions. PolyQ and polyQ/N tracts are structurally flexible sequences that trigger
protein aggregation. We report that due to their high percentages of STQ or STQN amino acid
content, four SCDs and three prion-causing Q/N-rich motifs of yeast proteins possess autonomous
protein expression-enhancing activities. Since these Q-rich motifs can endow proteins with structural
and functional plasticity, we suggest that they represent useful toolkits for evolutionary novelty.
Comparative Gene Ontology (GO) analyses of the near-complete proteomes of 26 representative
model eukaryotes reveal that Q-rich motifs prevail in proteins involved in specialized biological
processes, including *Saccharomyces cerevisiae* RNA-mediated transposition and pseudohyphal
growth, *Candida albicans* filamentous growth, ciliate peptidyl-glutamic acid modification and
microtubule-based movement, *Tetrahymena thermophila* xylan catabolism and meiosis, *Dictyos-
telium discoideum* development and sexual cycles, *Plasmodium falciparum* infection, and the
nervous systems of *Drosophila melanogaster, Mus musculus* and *Homo sapiens.* We also show that
Q-rich-motif proteins are expanded massively in 10 ciliates with reassigned $TAA^Q$ and $TAG^Q$ codons.
Notably, the usage frequency of $CAG^Q$ is much lower in ciliates with reassigned $TAA^Q$ and $TAG^Q$
codons than in organisms with expanded and unstable Q runs (e.g. *D. melanogaster* and *H. sapiens*),
indicating that the use of noncanonical stop codons in ciliates may have coevolved with codon
usage biases to avoid triplet repeat disorders mediated by CAG/GTC replication slippage.

## eLife assessment

This study presents **useful** results on glutamine-rich motifs in relation to protein expression and
alternative genetic codes. The **solid** data are based on bioinformatic approaches that are employed
to systematically uncover sequence features associated with proteome-wide amino acid distribution
and biological processes.

## Introduction

We reported previously that the $NH_2$-terminal domain (NTD; residues 1–66) of budding yeast *Saccha-
romyces cerevisiae* Rad51 protein contains three SQ motifs ($S^2Q$, $S^{12}Q$, and $S^{30}Q$; *Woo et al., 2020*).
The S/T-Q motifs, comprising S or T followed by Q, are the target sites of DNA damage sensor protein
kinases, that is ATM (ataxia-telangiectasia mutated), ATR (RAD3-related) (*Craven et al., 2002*; *Kim
et al., 1999*) and DNA-dependent protein kinase (DNA-PK) (*Traven and Heierhorst, 2005*; *Cheung*

*et al., 2012*). Mec1 (<u>M</u>itotic <u>E</u>ntry <u>C</u>heckpoint 1) and Tel1 (<u>TEL</u>omere maintenance 1) are the budding yeast homologs of mammalian ATR and ATM, respectively. Budding yeast lacks a DNA-PK homolog (*Craven et al., 2002*; *Kim et al., 1999*; *Menolfi and Zha, 2020*). This clustering of three SQ motifs within a stretch of 31 amino acids in Rad51-NTD fulfills the criteria to define an S/T-Q cluster domain (SCD; *Traven and Heierhorst, 2005*; *Cheung et al., 2012*). The three SQ motifs of Rad51-NTD are phosphorylated in a Mec1- and Tel1-dependent manner during vegetative growth and meiosis. Mec1/Tel1-dependent NTD phosphorylation antagonizes Rad51 degradation via the proteasomal pathway, increasing the half-life of Rad51 from 30 min to ≥180 min (*Woo et al., 2020*), supporting the notion that Mec1 and Tel1 exhibit an essential function in regulating protein homeostasis (proteostasis) in *S. cerevisiae* (*Corcoles-Saez et al., 2018*; *Corcoles-Saez et al., 2019*).

A unifying definition of an SCD is having ≥3 S/T-Q sites within a stretch of 50–100 amino acids (*Traven and Heierhorst, 2005*; *Cheung et al., 2012*). One of the best-understood mechanisms of SCD phosphorylation involves the association of SCDs with their binding partners containing a forkhead-associated (FHA) domain. For example, Mec1/Tel1-dependent phosphorylation of Rad53-SCD1 (residues 1–29) and Hop1-SCD (residues 258–324) specifically recruits and activates their downstream DNA damage checkpoint kinases Dun1 and Mek1, respectively (*Lee et al., 2008*; *Carballo et al., 2008*; *Chuang et al., 2012*). Dun1 phosphorylates three serine residues (S56, S58, and S60) of the ribonucleotide reductase inhibitor Sml1, subsequently promoting Sml1 ubiquitination by the E2 ubiquitin-conjugating enzyme Rad6 and the E3 ubiquitin ligase Ubr2, as well as promoting Sml1 degradation via the 26 S proteasome (*Zhao and Rothstein, 2002*; *Uchiki et al., 2004*; *Andreson et al., 2010*). Mek1 phosphorylates two Rad51 accessory factors, Rad54 and Hed1 (a meiosis-specific inhibitor of Rad51), suppressing Rad51's strand-exchange activity and preventing Rad51-mediated DSB repair, respectively (*Niu et al., 2009*; *Callender et al., 2016*).

There are many other SCD-containing proteins that are neither targets of ATM/Tel1 or ATR/Mec1 nor functionally linked to DNA Damage response or DNA repair (*Traven and Heierhorst, 2005*; *Cheung et al., 2012*), indicating that SCDs might possess previously uncharacterized biochemical properties or physiological functions. Interestingly, due to their high percentages of STQ amino acid content, SCDs often are intrinsically disordered regions (IDRs) in their native states rather than adopting stable secondary and/or tertiary structures (*Traven and Heierhorst, 2005*). A common feature of IDRs is their high content of serine (S), threonine (T), glutamine (Q), asparanine (N), proline (P), glycine (G) or charged amino acids [arginine (R), lysine (K), and histidine (H)] (*Romero et al., 2001*; *Uversky, 2019*; *Macossay-Castillo et al., 2019*; *Uversky et al., 2000*). Functionally, IDRs are key components of subcellular machineries and signaling pathways because they have the potential to associate with many partners due to their multiple possible metastable conformations. Many IDRs are regulated by alternative splicing and post-translational modifications. Some IDRs are involved in the formation of various membraneless organelles via intracellular liquid-liquid phase separation (*Wright and Dyson, 1999*; *Posey et al., 2018*). Highly charged IDRs can act as entropic bristles that, when translationally fused to their partner proteins, only enhance the water solubility but not the steady-state levels of their partner proteins (*Santner et al., 2012*).

In this study, we first report that seven Q-rich motifs of *S. cerevisiae* proteins, including Rad51-NTD (*Woo et al., 2020*), have high STQN or STQ amino acid contents and exhibit autonomous expression-enhancing activity for high-level production of native protein and when fused to exogenous target proteins, for example **β-**galactosidase (LacZ), in vivo. We also reveal structural and genetic requirements for the 'nanny' function of these Q-rich motifs in regulating protein homeostasis, leading to the hypothesis that Q-rich motifs are useful toolkits for structural and functional plasticity, as well as evolutionary novelty. Next, we performed Gene Ontology (GO) enrichment analyses on all proteins having Q-rich motifs (i.e. SCDs, polyQ and polyQ/N), as well as those with the homorepeat (polyX) motifs of other amino acid residues, in 20 non-ciliate and 17 ciliate species. Notably, relative to most other eukaryotes, many ciliates reassign their standard stop codons into amino acids (*Table 1*). For example, several ciliates *possess two* noncanonical nuclear genetic codes (UAA$^Q$ and UAG$^Q$), in which the UAA and UAG stop codons have been reassigned to glutamine (Q) so that UGA is the sole functional stop codon, including *Tetrahymena thermophila*, *Paramecium tetraurelia*, *Paramecium bursaria*, *Oxytricha trifallax*, *Stylonychia lemnae*, *Pseudocohnilembus persalinus*, *Aristerostoma* sp., *Favella ehrenbergii*, *Pseudokeronopsis* spp., *Strombidium inclinatum,* and *Uronema* spp. Both the UAA and UAG stop codons are reassigned to tyrosine (Y) in *Favella ehrenbergii*, whereas the UGA stop codon is translated

**Table 1.** Usage frequency (%) of standard codons [stop codon (*), Q, C, Y and W] and reassigned stop codons (→ Q, → C or → W) in 37 different eukaryotes.

| Species | Source | ID | BUSCO Protein (%) | Protein # | TAA | TAG | TGA | CAA | CAG | TGC | TGT | TAC | TAT | TGG |
|---|---|---|---|---|---|---|---|---|---|---|---|---|---|---|
| **NCBI genetic code: 1** | **Non-ciliate eukaryotes** | | | | * | * | * | Q | Q | C | C | Y | Y | W |
| Saccharomyces cerevisiae | UniProt | UP000002311 | 99.6 | 6062 | 0.16 | 0.08 | 0.01 | 2.77 | 0.89 | 0.63 | 1.03 | 0.86 | 3.10 | 0.93 |
| Candida albicans | UniProt | UP000000559 | 98.8 | 6035 | 0.10 | 0.05 | 0.03 | 3.57 | 0.65 | 0.18 | 0.94 | 1.04 | 2.54 | 1.09 |
| Candida auris | UniProt | UP000230249 | 97.4 | 5409 | 0.08 | 0.06 | 0.06 | 1.81 | 2.12 | 0.55 | 0.59 | 2.09 | 1.16 | 1.07 |
| Candida tropicalis | UniProt | UP000002037 | 94.6 | 6226 | 0.10 | 0.07 | 0.03 | 3.61 | 0.66 | 0.14 | 0.96 | 0.95 | 2.62 | 0.98 |
| Neurospora crassa | UniProt | UP000001805 | 99.2 | 10257 | 0.06 | 0.05 | 0.08 | 1.70 | 2.60 | 0.77 | 0.34 | 1.75 | 0.85 | 1.31 |
| Magnaporthe oryzae | UniProt | UP000009058 | 98.6 | 12794 | 0.06 | 0.07 | 0.10 | 1.37 | 2.69 | 0.92 | 0.35 | 1.80 | 0.71 | 1.42 |
| Trichoderma reesei | PMID: 34908505 | PRJNA382020 | 99.2 | 13735 | 0.06 | 0.06 | 0.11 | 1.17 | 2.95 | 0.95 | 0.32 | 1.80 | 0.83 | 1.42 |
| Cryptococcus neoformans | UniProt | UP000002149 | 99.5 | 6743 | 0.07 | 0.06 | 0.05 | 2.06 | 1.79 | 0.48 | 0.55 | 1.39 | 1.14 | 1.37 |
| Ustilago maydis | UniProt | UP000000561 | 99.4 | 6806 | 0.04 | 0.05 | 0.07 | 1.82 | 2.61 | 0.72 | 0.35 | 1.59 | 0.65 | 1.18 |
| Taiwanofungus camphoratus | PMID: 35196809 | PRJNA615295 | 94.6 | 14019 | 0.05 | 0.06 | 0.11 | 1.57 | 2.19 | 0.70 | 0.57 | 1.38 | 1.22 | 1.36 |
| Dictyostelium discoideum | UniProt | UP000002195 | 93.7 | 12734 | 0.16 | 0.01 | 0.01 | 4.86 | 0.19 | 0.15 | 1.27 | 0.52 | 3.02 | 0.73 |
| Plasmodium falciparum | UniProt | UP000001450 | 99.1 | 5376 | 0.09 | 0.01 | 0.03 | 2.42 | 0.37 | 0.23 | 1.52 | 0.61 | 5.05 | 0.49 |
| Drosophila melanogaster | UniProt | UP000000803 | 100 | 22088 | 0.08 | 0.07 | 0.05 | 1.56 | 3.61 | 1.32 | 0.54 | 1.84 | 1.08 | 0.99 |
| Aedes aegypti | UniProt | UP000008820 | 99.4 | 18998 | 0.11 | 0.07 | 0.08 | 1.76 | 2.58 | 1.11 | 0.79 | 2.16 | 1.14 | 1.06 |
| Caenorhabditis elegans | UniProt | UP000001940 | 100 | 26548 | 0.16 | 0.06 | 0.14 | 2.74 | 1.44 | 0.91 | 1.12 | 1.37 | 1.75 | 1.11 |
| Danio rerio | UniProt | UP000000437 | 95.5 | 46844 | 0.11 | 0.06 | 0.14 | 1.18 | 3.35 | 1.12 | 1.13 | 1.70 | 1.26 | 1.16 |
| Mus musculus | UniProt | UP000000589 | 99.7 | 55341 | 0.10 | 0.08 | 0.16 | 1.20 | 3.41 | 1.23 | 1.14 | 1.61 | 1.22 | 1.25 |
| Homo sapiens | UniProt | UP000005640 | 99.5 | 79038 | 0.10 | 0.08 | 0.16 | 1.23 | 3.42 | 1.26 | 1.06 | 1.53 | 1.22 | 1.32 |
| Arabidopsis thaliana | UniProt | UP000006548 | 100 | 39334 | 0.09 | 0.05 | 0.12 | 1.94 | 1.52 | 0.72 | 1.05 | 1.37 | 1.46 | 1.25 |
| Chlamydomonas reinhardtii | UniProt | UP000006906 | 98.9 | 18829 | 0.03 | 0.04 | 0.06 | 0.59 | 4.05 | 1.1 | 0.22 | 1.45 | 0.24 | 1.16 |
| **NCBI genetic code: 6** | **group I ciliates** | | | | → Q | → Q | * | Q | Q | C | C | Y | Y | W |
| Tetrahymena thermophila | UniProt | UP000009168 | 98.9 | 26972 | 5.46 | 1.63 | 0.16 | 2.04 | 0.48 | 0.79 | 0.99 | 1.22 | 3.09 | 0.51 |

*Table 1 continued on next page*

Table 1 continued

| Species | Source | ID | BUSCO Protein (%) | Protein # | TAA | TAG | TGA | CAA | CAG | TGC | TGT | TAC | TAT | TGG |
|---|---|---|---|---|---|---|---|---|---|---|---|---|---|---|
| Paramecium tetraurelia | UniProt | UP000000600 | 98.8 | 39461 | 4.53 | 1.48 | 0.22 | 2.54 | 0.57 | 0.61 | 1.21 | 1.12 | 3.14 | 0.76 |
| Oxytricha trifallax | UniProt | UP000006077 | 97.1 | 23559 | 3.63 | 1.57 | 0.15 | 2.68 | 1.07 | 0.59 | 0.56 | 1.44 | 2.27 | 0.58 |
| Stylonychia lemnae | UniProt | UP000039865 | 97.1 | 20720 | 3.22 | 1.81 | 0.17 | 2.26 | 1.05 | 0.62 | 0.55 | 1.31 | 2.49 | 0.62 |
| Pseudocohnilembus persalinus | UniProt | UP000054937 | 92.4 | 13175 | 7.36 | 1.39 | 0.18 | 1.76 | 0.37 | 0.32 | 1.00 | 1.00 | 3.26 | 0.61 |
| NCBI genetic code: 6 | group II ciliates | | | | →Q | →Q | * | Q | Q | C | C | Y | Y | W |
| Aristerostoma | MMETSP | MMETSP0125 | 62.5 | 27868 | 0.96 | 1.04 | 0.15 | 2.65 | 0.97 | 0.71 | 0.68 | 1.35 | 2.49 | 0.8 |
| Favella ehrenbergii | MMETSP | MMETSP0123 | 85.4 | 26477 | 0.72 | 1.51 | 0.16 | 1.88 | 3.06 | 1.11 | 0.25 | 2.06 | 0.71 | 0.83 |
| Pseudokeronopsis | MMETSP | MMETSP0211 MMETSP1396 | 87.2 | 62574 | 1.04 | 1.37 | 0.16 | 2.05 | 2.58 | 0.94 | 0.44 | 2.18 | 1.40 | 0.78 |
| Strombidium inclinatum | MMETSP | MMETSP0208 | 83.6 | 32210 | 0.64 | 1.28 | 0.11 | 1.63 | 3.50 | 0.83 | 0.24 | 2.12 | 0.69 | 0.7 |
| Uronema spp. | MMETSP | MMETSP0018 | 52.6 | 13887 | 6.90 | 0.66 | 0.17 | 0.80 | 0.08 | 0.28 | 1.63 | 0.80 | 3.62 | 0.87 |
| NCBI genetic code: 1 | group III ciliates | | | | * | * | * | Q | Q | C | C | Y | Y | W |
| Stentor coeruleus | UniProt | UP000187209 | 92.4 | 30969 | 0.16 | 0.08 | 0.01 | 2.77 | 0.89 | 0.63 | 1.03 | 0.86 | 3.1 | 0.93 |
| Climacostomum virens | MMETSP | MMETSP1397 | 94.7 | 33899 | 0.11 | 0.09 | 0.04 | 1.79 | 2.20 | 1.38 | 0.60 | 2.60 | 0.85 | 1.06 |
| Litonotus pictus | MMETSP | MMETSP0209 | 65.5 | 30222 | 0.08 | 0.03 | 0.01 | 2.12 | 1.52 | 0.63 | 0.77 | 1.83 | 2.25 | 0.54 |
| Protocruzia adherens | MMETSP | MMETSP0216 | 74.9 | 40577 | 0.07 | 0.04 | 0.04 | 2.91 | 1.24 | 0.69 | 0.94 | 1.30 | 1.83 | 1.00 |
| NCBI genetic code: 10 | group IV ciliate | | | | * | * | →C | Q | Q | C | C | Y | Y | W |
| Euplotes focardii | MMETSP | MMETSP0205 MMETSP0206 | 60.8 | 36659 | 0.23 | 0.06 | 0.51 | 2.43 | 1.23 | 0.49 | 0.84 | 1.28 | 2.38 | 0.87 |
| NCBI genetic code: 4 | group IV ciliate | | | | * | * | →W | Q | Q | C | C | Y | Y | W |
| Blepharisma japonicum | MMETSP | MMETSP1395 | 81.9 | 22714 | 0.13 | 0.03 | 0.30 | 2.85 | 1.24 | 0.94 | 0.80 | 0.94 | 2.72 | 0.84 |
| NCBI genetic code: 29 | group IV ciliate | | | | →Y | →Y | * | Q | Q | C | C | Y | Y | W |
| Mesodinium pulex | MMETSP | MMETSP0467 | 88.9 | 61058 | 0.29 | 0.56 | 0.13 | 0.77 | 3.33 | 1.53 | 0.25 | 1.78 | 0.34 | 1.29 |

to tryptophan (W) or cysteine (C) in *Blepharisma japonicum* and *Euplotes focardii*, respectively. In contrast, *Stentor coeruleus*, *Climacostomum virens*, *Litonotus pictus* and *Protocruzia adherens* utilize the universal set of genetic codons. *Condylostoma magnum* and *Parduczia* sp. have no dedicated genetic codes. Their UAA, UAG and UGA codons can be stop codons or translated to Q, C, and W, respectively. Translation termination at the mRNA 3′ end occurs in a context-dependent manner to distinguish stop from sense (*Caron and Meyer, 1985*; *Helftenbein, 1985*; *Horowitz and Gorovsky, 1985*; *Preer et al., 1985*; *Lozupone et al., 2001*; *Ring and Cavalcanti, 2008*; *Salim et al., 2008*; *Dohra et al., 2015*; *Xiong et al., 2015*; *Swart et al., 2016*; *Heaphy et al., 2016*; *Slabodnick et al., 2017*; *Kollmar and Mühlhausen, 2017*). Although it has been reported previously that Q is used more frequently in *Tetrahymena thermophila* and *Paramecium tetraurelia* than in other species (*Ring and Cavalcanti, 2008*; *Salim et al., 2008*), many important questions regarding stop codon reassignment in ciliates remain unresolved. For instance, fundamentally, it is unclear if Q, Y, W, and C are used more frequently in other ciliates in which stop codons are reassigned. Moreover, whether there are common or specific structural motif(s) in proteins arising from stop codon reassignment is not clear. Furthermore, what are the structural and functional impacts of such genome-wide alterations? Finally, the codons that code for the polyQ motifs are prone to a CAG/GTC-slippage mechanism during DNA replication, and thus many Q-runs are unstable and expanded in some eukaryotic organisms, leading to polyQ-associated diseases (e.g. Huntington's disease; *Petruska et al., 1998*; *Ruff et al., 2017*; *Mier and Andrade-Navarro, 2021*). Accordingly, polyQ tracts in proteins associated with disease are more enriched in the CAG codon, becoming almost CAG exclusive (*Mier and Andrade-Navarro, 2021*; *Nalavade et al., 2013*). In contrast, CAA insertions in a *Drosophila* model of a polyQ-associated disease revealed that even though the resulting polyQ tract is of the same length as the disease-associated tract, the proteins display reduced toxicity (*Li et al., 2008*). In this study, we determine and compare the usage frequency of TAA$^Q$, TAG$^Q$, CAA$^Q$, and CAG$^Q$ in ciliates and non-ciliate eukaryotes.

## Results

### SCDs provide versatile functionalities in proteins

We have shown previously in *S. cerevisiae* that Rad51-NTD autonomously promotes high-level production of native Rad51 and its COOH-terminal fusion protein LacZ (β-galactosidase) in vivo (*Woo et al., 2020*). To do so, in brief, we expressed Rad51-NTD-LacZ-NVH fusion proteins using a *CEN-ARS* plasmid (low-copy number) under the control of the native *RAD51* gene promoter ($P_{RAD51}$) (*Table 1*). The NVH tag contains an SV40 nuclear localization signal (<u>N</u>LS) peptide preceding a <u>V</u>5 epitope tag and a hexahistidine (<u>His</u>$_6$) affinity tag (*Woo et al., 2020*). We confirmed that the N-terminal addition of Rad51-NTD to LacZ-NVH increased both steady-state levels of LacZ-NVH fusion proteins (*Figure 1A*) and β-galactosidase activities in vivo (*Figure 1B*). Here, we further report that yeast Rad53-SCD1, Hop1-SCD, Sml1-NTD$^{1-50}$ (residues 1–50) and Sml1-NTD$^{1-27}$ (residues 1–27) also exhibit protein expression-enhancing (PEE) activities (*Figure 1A and B*, *Table 1*). The Sml1 protein in the SK1 strain harbors three S/T-Q motifs (S$^4$Q, S$^{14}$Q and T$^{47}$Q), whereas that in the S288c strain only has one SQ motif (S$^4$Q, C$^{14}$Q and T$^{47}$M).

### The Q-rich motifs of three yeast prion-causing proteins also exhibit PEE activities

Since Sml1-NTD$^{1-27}$ in the SK1 strain only harbors two S/T-Q motifs (S$^4$Q and S$^{14}$Q), the number of S/T-Q motifs alone could not account for PEE activity. Notably, Rad51-NTD, Rad53-SCD1, Hop1-SCD, Sml1-NTD$^{1-27}$ and Sml1-NTD$^{1-50}$ all represent Q- or Q/N-rich motifs. Rad51-NTD contains 9 serines (S), 2 threonines (T), 9 glutamines (Q), and 4 asparagines (N). Rad53-SCD1 has 2 S, 4T, 7Q, and 1 N. Hop1-SCD has 6 S, 6T, 8Q, and 9 N. Sml1-NTD$^{1-27}$ and Sml1-NTD$^{1-50}$ in SK1 possess 3 S and 5 S, 2T and 3T, 6Q and 7Q, as well as 2 N and 3 N, respectively.

Accordingly, we investigated if other Q- or Q/N-rich motifs in yeast can also promote protein expression in vivo. PolyQ and polyQ/N tracts are the most common homorepeats acting as structurally flexible motifs for protein aggregation or protein–protein interactions in many eukaryotes (*Chavali et al., 2017*; *Mier et al., 2020*). PolyN is not as structurally flexible as polyQ due to a stronger propensity for β-turn formation in polyN than in polyQ (*Lu and Murphy, 2014*). In so-called polyQ-associated diseases, long Q-, Q/N- or even N-rich motifs cause an excess of interactions, resulting in dysfunctional

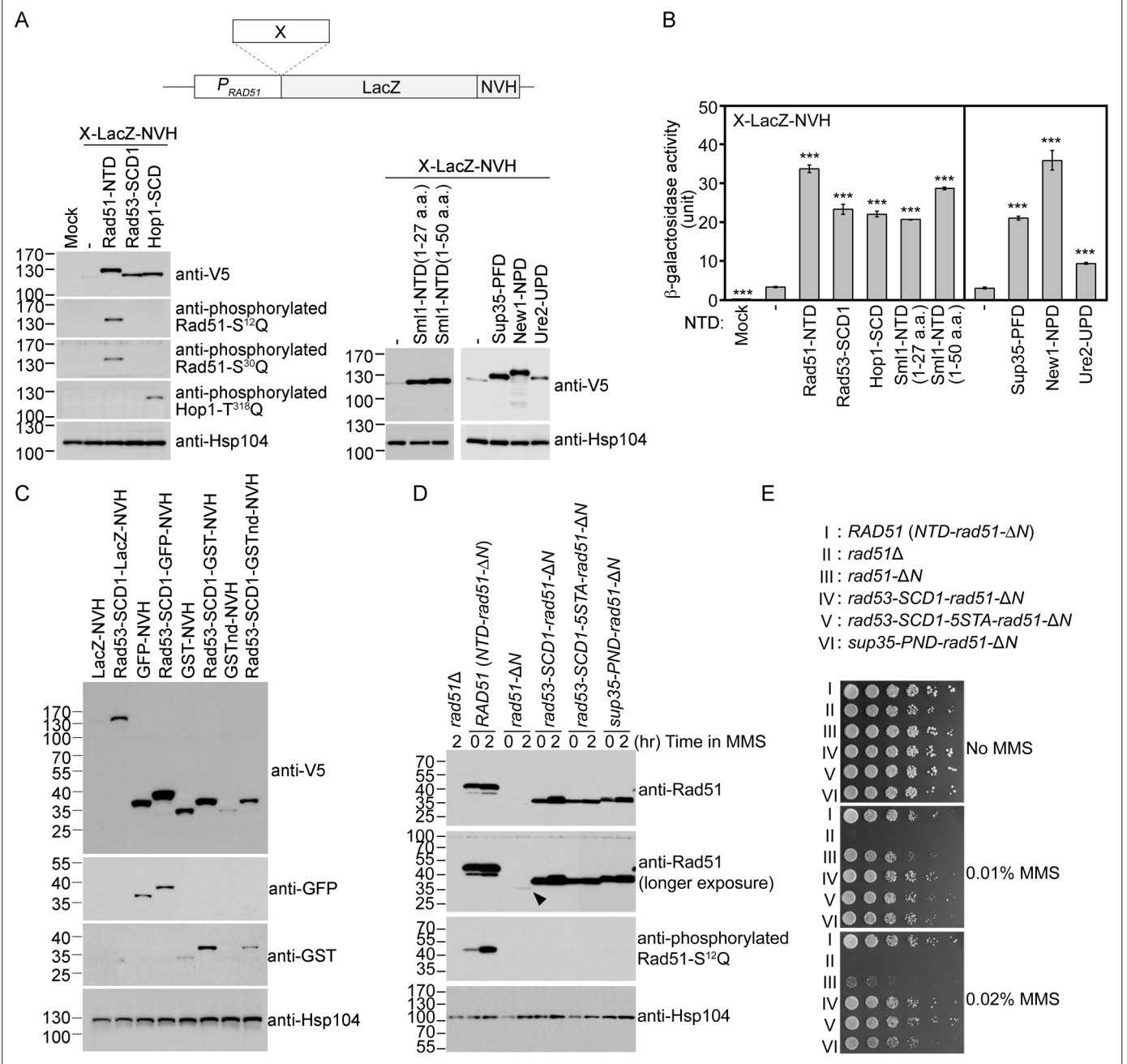

**Figure 1.** The Q-rich domains of seven different yeast proteins possess autonomous expression-enhancing (PEE) activities. (**A–B**) N-terminal fusion of Rad51-NTD/SCD, Rad53-SCD1, Hop1-SCD, Sml1-NTD, Sup35-PND, Ure2-UPD and New1-NPD promotes high-level expression of LacZ-NVH, respectively. The NVH tag contains an SV40 nuclear localization signal (NLS) peptide preceding a V5 epitope tag and a hexahistidine (His₆) affinity tag (*Woo et al., 2020*). Western blots for visualization of LacZ-NVH fusion proteins (**A**) and quantitative β-galactosidase assays (**B**) were carried out as described previously (*Woo et al., 2020*). Error bars indicate standard deviation between experiments (n≥3). Asterisks indicate significant differences relative to wild type (WT) in A or lacking an NTD in B, with p values calculated using a two-tailed *t*-test (***, p-value <0.001; **, p-value <0.01). (**C–D**) The PEE activities of S/T/Q/N-rich domains are independent of the quaternary structures of target proteins. (**C**) Rad53-SCD1 can be used as an N-terminal fusion tag to enhance production of four different target proteins: LacZ-NVH, GST-NVH, GSTnd-NVH, and GFP-NVH. (**D**) Visualization of native Rad51 (NTD-Rad51-ΔN), Rad51-ΔN, and the Rad51-ΔN fusion proteins by immunoblotting. Hsp104 was used as a loading control. Size in kilodaltons of standard protein markers is labeled to the left of the blots. The black arrowhead indicates the protein band of Rad51-ΔN. (**E**) MMS sensitivity. Spot assay showing fivefold serial dilutions of indicated strains grown on YPD plates with or without MMS at the indicated concentrations (w/v).

The online version of this article includes the following source data for figure 1:

**Source data 1.** Raw and labelled images for blots shown in *Figure 1*.

or pathogenic protein aggregates (*Zoghbi and Orr, 2000*). Many prion-causing proteins contain Q/N-rich prion-forming domains (PFDs). In *S. cerevisiae*, the best-characterized prion-causing proteins are Sup35 (or translation terminator eRF35), New1 ([*NU+*] prion formation protein 1), Ure2 (uridosuccinate transport 2), Rnq1 (rich in N and Q 1), and Swi1 (switching deficient 1) (*Michelitsch and Weissman, 2000*; *Uptain and Lindquist, 2002*). We found that the Q/N-rich NTDs of Sup35, Ure2 and New1 also display PEE activities, i.e., the prion nucleation domain (PND; residues 1–39) of Sup35 (*Tuite, 2000*), the Ure2 prion domain (UPD) (residues 1–91) (*Wickner et al., 2004*; *Wickner, 1994*), and the New1 prion domain (NPD; residue 1–146) (*Shewmaker et al., 2007*; *Figure 1A and B*, *Supplementary file 1a*). Sup35-PND containing 3 S, 12Q, 18 N, and an $S^{17}Q$ motif exerts critical functions in promoting [*PSI+*] prion nucleation (*Toombs et al., 2011*). The UPD of the Ure2 nitrogen catabolite repression transcriptional regulator is the basis of the prion [*URE3⁺*] (*Wickner et al., 2004*; *Wickner, 1994*). The UPD is critical for Ure2's function in vivo because its removal in the corresponding Ure2-ΔUPD mutants elicits reduced protein stability and steady-state protein levels (but not transcript levels) (*Shewmaker et al., 2007*). Ure2-UPD contains 10 S, 5T, 10Q, and 33 N, adopting a completely disordered structure (*Ngo et al., 2012*). New1 is a non-essential ATP-binding cassette type F protein that fine-tunes the efficiency of translation termination or ribosome recycling (*Kasari et al., 2019*). The NPD of New1 supports [*NU⁺*] and is susceptible to [*PSI+*] prion induction (*Santoso et al., 2000*; *Osherovich and Weissman, 2001*). New1-NPD contains 19 S, 8T, 14Q, 28 N and an $S^{145}Q$ motif. Here, we applied the LacZ-NVH fusion protein approach to show that N-terminal fusion of Sup35-PND, Ure2-UPD or New1-NPD to LacZ-NVH all increased steady-state protein levels (*Figure 1A*) and β-galactosidase activities in vivo (*Figure 1B*).

## The PEE function is not affected by the quaternary structures of target proteins

We found that N-terminal fusion of Rad53-SCD1 to four different NVH-tagged target proteins (*Figure 1C*) or Rad51-ΔN (*Figure 1D*) all resulted in higher protein production in vivo. LacZ is a tetrameric protein, glutathione S-transferase (GST) is dimeric, and non-dimerizing GST (GSTnd) and GFP are monomeric proteins. As reported recently (*Woo et al., 2020*), removal of the NTD from Rad51 reduced by ~97% the levels of corresponding Rad51-ΔN proteins relative to wild type (WT) (*Figure 1D*), leading to lower resistance to the DNA damage agent methyl methanesulfonate (MMS) (*Figure 1E*). Interestingly, the autonomous PEE function of Rad51-NTD could be fully rescued in *rad51-ΔN* (*Supplementary file 1*) by N-terminal fusion of Rad53-SCD1, Rad53-SCD1-5STA (all five S/T-Q motifs changed to AQs) or Sup35-PND, respectively. Rad53-SCD1-5STA is a mutant protein defective in Mec1- and Tel1-mediated phosphorylation. Compared to WT yeast cells, the three corresponding yeast mutants (*rad53-SCD1-rad51-ΔN*, *rad53-SCD1-5STA-rad51-ΔN* and *sup35-PND-rad51-ΔN*) not only produced similar steady-state levels of Rad51-ΔN fusion proteins (*Figure 1D*), but they also exhibited high MMS resistance (*Figure 1E*).

During homology-directed repair of DNA double-strand breaks (DSBs), Rad51 polymerizes into helical filaments on DSB-associated single-stranded DNA (ssDNA) and then promotes homologous search and strand exchange of the ssDNA-protein filament with a second double-stranded DNA (dsDNA). We inferred that the catalytic activity of Rad51-ΔN during DSB repair is likely similar to that of wild-type Rad51 because the weak MMS-resistant phenotype of *rad51-ΔN* is mainly due to very low steady-state levels of Rad51-ΔN (*Figure 1D*).

In conclusion, our results indicate that the quaternary structures of the target proteins (i.e. GFP, GSTnd, GST, LacZ and Rad51-ΔN) are irrelevant to the autonomous PEE activity. We assert that our use of a nuclear localization signal on the C-terminal VHN tag was unlikely to influence protein degradation kinetics or to sequester the reporter, leading to their accumulation and the appearance of enhanced expression for two reasons. First, the negative control LacZ-NV also possesses the same nuclear localization signal (*Figure 1A*, lane 2). Second, as an endogenous fusion target, Rad51-ΔN does not harbor the NVH tag (*Figure 1D*, lanes 3–4). Compared to WT Rad51, Rad51-ΔN is highly labile. In our previous study, removal of the NTD from Rad51 reduced by ~97% the protein levels of corresponding Rad51-ΔN proteins relative to WT (*Woo et al., 2020*).

## The autonomous PEE function is not likely controlled by plasmid copy number or its transcription

The PEE function is unlikely to operate at the transcriptional level, as revealed by genomic and *reverse-transcription quantitative polymerase chain reaction* analyses (i.e. g-qPCR and RT-qPCR, respectively)

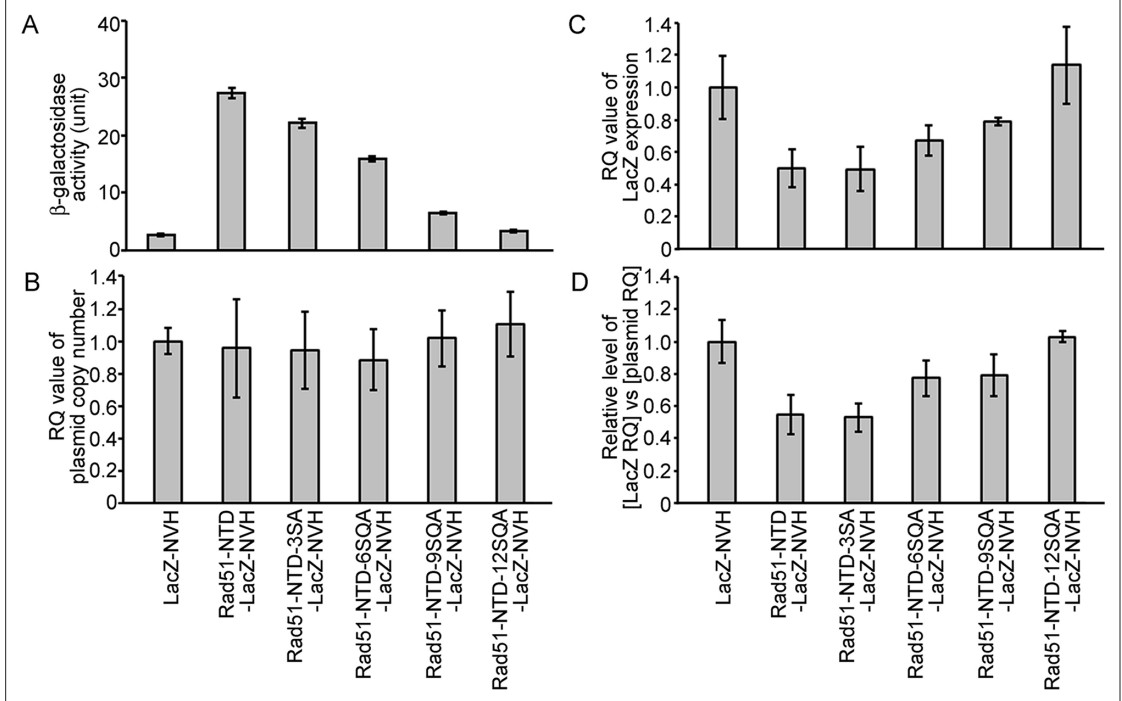

**Figure 2.** The autonomous protein-expression-enhancing function of Rad51-NTD is unlikely to be controlled during transcription or simply arise from plasmid copy number differences. The effects of WT and mutant Rad51-NTD on β-galactosidase activities (**A**), plasmid DNA copy numbers (**B**), relative steady-state levels of LacZ-NVH mRNA normalized to *ACT1* (actin) mRNA (**C**), and relative ratios of LacZ-NVH mRNA *versus* plasmid DNA copy number (**D**). The wild-type yeast cells were transformed with indicated CEN-ARS plasmids, respectively, to express WT and mutant Rad51-NTD-LacZ-NVH fusion proteins or LacZ-NVH alone under the control of the native RAD51 gene promoter (P$_{RAD51}$). The relative quantification (RQ = $2^{-\Delta\Delta C}$T) values were determined to reveal the plasmid DNA copy number and steady-state levels of LacZ-NVH mRNA by g-qPCR and RT-qPCR, respectively. LacZ and *ACT1* were selected as target and reference protein-encoding genes, respectively, in both g-qPCR and RT-qPCR. The data shown represent mean ± SD from three independent biological data-points.

The online version of this article includes the following source data for figure 2:

**Source data 1.** The raw qPCR data of cDNA and gDNA in *Figure 2*.

(*Figure 2*, *Supplementary file 1c*, and *Figure 2—source data 1*). We found that the addition of WT and mutant Rad51-NTD to LacZ-NVH not only did not affect the average copy number of the corresponding *CEN-ARS* plasmids in exponentially growing *S. cerevisiae* cells (*Figure 2A*), but also even reduced the steady-state transcript levels of the corresponding LacZ-NVH fusion protein genes (*Figure 2B*). Therefore, the addition of Rad51-NTD to LacZ-NVH did not result in a significant increase in transcription.

## The protein quality control system moderately regulates autonomous PEE activities

The protein quality control system is a mechanism by which cells monitor proteins to ensure that they are appropriately folded (*Chen et al., 2011*). In the current study, we compared the protein steady-state levels (*Figure 3A*) and β-galactosidase activities (*Figure 3B–D*) of Rad51-NTD-LacZ-NVH and LacZ-NVH in WT, *hsp104Δ*, *new1Δ*, *doa1Δ*, *doa4Δ*, *san1Δ* and *oaz1Δ* yeast cell lines. The protein products encoded by each of the six genes deleted from the latter mutant lines are all functionally relevant to protein homeostasis or prion propagation. Hsp104 is a heat-shock protein with disaggregase activities that disrupts protein aggregation (*Shorter and Southworth, 2019*; *Ye et al., 2020*). New1 is a translation factor that fine-tunes ribosome recycling and the efficiency of translation termination (*Kasari et al., 2019*). Doa1 (also called Ufd3) is an ubiquitin- and Cdc48-binding protein with a role in ubiquitin homeostasis and/or protein degradation (*Mullally et al., 2006*; *Zhao et al., 2009*). The *doa1Δ* mutant exhibits diminished formation of [*PSI+*] prion (*Tyedmers et al., 2008*). Doa4 is a deubiquitinating enzyme required for recycling ubiquitin from proteasome-bound ubiquitinated

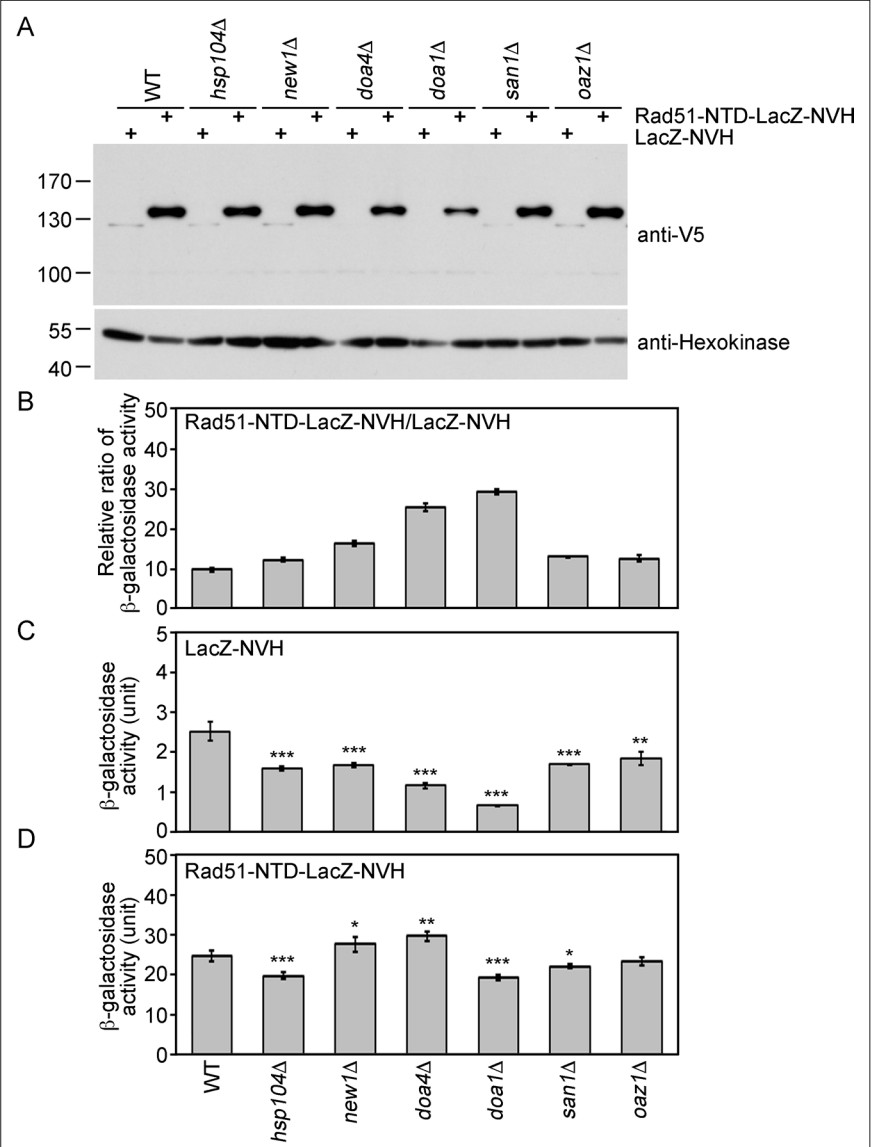

**Figure 3.** The expression-promoting function of Rad51-NTD is controlled during protein translation and does not affect ubiquitin-mediated protein degradation. (**A**) The steady-state protein levels of Rad51-NTD-LacZ-NVH and LacZ-NVH in WT and six protein homeostasis gene knockout mutants. (**B–D**) The impact of six protein homeostasis genes on the β-galactosidase activity ratios of Rad51-NTD-LacZ-NVH to LacZ-NVH in WT and the six gene knockout mutants (**B**). The β-galactosidase activities of LacZ-NVH (**C**) and Rad51-NTD-LacZ-NVH (**D**) in WT and the six gene knockout mutants are shown. Asterisks indicate significant differences, with values calculated using a two-tailed $t$-test (***, p-value <0.001; **, p-value <0.01; *, p-value <0.05).

The online version of this article includes the following source data for figure 3:

**Source data 1.** Raw and labelled images for blots shown in **Figure 3**.

intermediates (**Swaminathan et al., 1999**). The *doa4Δ* mutant exhibits increased sensitivity to the protein synthesis inhibitor cycloheximide (**Dudley et al., 2005**). San1 is an ubiquitin-protein ligase that targets highly aggregation-prone proteins (**Dasgupta et al., 2004**; **Fredrickson et al., 2013**). Oaz1 (ornithine decarboxylase antizyme) stimulates ubiquitin-independent degradation of Spe1 ornithine decarboxylase by the proteasome (**Porat et al., 2008**). We found that the β-galactosidase activities of Rad51-NTD-LacZ-NVH in WT and all six of the gene-knockout strains we examined were 10- to 29-fold higher than those of LacZ-NVH (**Figure 3B**). Intriguingly, the β-galactosidase activities of LacZ-NVH in the six gene-knockout mutants are all lower (30–70%) than those in WT (**Figure 3C**). In contrast, the

β-galactosidase activities of Rad51-NTD-LacZ-NVH in WT are either slightly higher or lower than those in the six null mutants (*Figure 3D*). These results indicate that the addition of Rad51-NTD to LacZ-NVH can abrogate the protein homeostasis defects caused by the loss of each of these six genes. For example, Rad51-NTD might compensate for the ribosome assembly and translation defects in *new1Δ* (*Kasari et al., 2019*), as well as the cycloheximide-hypersensitive phenotype displayed by *doa4Δ* (*Dudley et al., 2005*). Accordingly, the β-galactosidase activities of Rad51-NTD-LacZ-NVH in the *new1Δ* and *doa4Δ* lines are higher than those in the WT, respectively. In contrast, the β-galactosidase activities of LacZ-NVH in the *new1Δ* and *doa4Δ* lines are lower, respectively, than those of WT. Finally, although the *doa1Δ* mutant is defective in [*PSI+*] prion formation (*Tyedmers et al., 2008*), the steady-state levels of Rad51-NTD-LacZ-NVH in the *doa1Δ* line are also slightly higher than those in WT.

## The N-end rule is not likely relevant to the PEE function of Q-rich motifs

The N-end rule links the in vivo half-life of a protein to the identity of its N-terminal residues. In *S. cerevisiae*, the N-end rule operates as part of the ubiquitin system and comprises two pathways. First, the Arg/N-end rule pathway, involving a single N-terminal amidohydrolase Nta1, mediates deamidation of N-terminal asparagine (N) and glutamine (Q) into aspartate (D) and glutamate (E), which in turn are arginylated by a single Ate1 R-transferase, generating the Arg/N degron. N-terminal R and other primary degrons are recognized by a single N-recognin Ubr1 in concert with ubiquitin-conjugating Ubc2/Rad6. Ubr1 can also recognize several other N-terminal residues, including lysine (K), histidine (H), phenylalanine (F), tryptophan (W), leucine (L), and isoleucine (I) (*Bachmair et al., 1986*; *Tasaki et al., 2012*; *Varshavsky, 2019*). Second, the Ac/N-end rule pathway targets proteins containing N-terminally acetylated (Ac) residues. Prior to acetylation, the first amino acid methionine (M) is catalytically removed by Met-aminopeptidases (MetAPs), unless a residue at position 2 is non-permissive (too large) for MetAPs. If a retained N-terminal M or otherwise a valine (V), cysteine (C), alanine (A), serine (S) or threonine (T) residue is followed by residues that allow N-terminal acetylation, the proteins containing these AcN degrons are targeted for ubiquitylation and proteasome-mediated degradation by the Doa10 E3 ligase (*Hwang et al., 2010*).

For two reasons, the PEE activities of these Q-rich domains are unlikely to arise from counteracting the N-end rule. First, the first two amino acid residues of Rad51-NTD, Hop1-SCD, Rad53-SCD1, Sup35-PND, Rad51-ΔN, and LacZ-NVH are MS, ME, ME, MS, ME, and MI, respectively, where M is methionine, S is serine, E is glutamic acid and I is isoleucine. Second, Sml1-NTD behaves similarly to these N-terminal fusion tags, despite its methionine and glutamine (MQ) amino acid signature at the N-terminus.

## The relationship between PEE function, amino acid contents and structural flexibility

We applied an alanine scanning mutagenesis approach to reduce the percentages of S, T, Q, or N in Rad51-NTD, Rad53-SCD1, and Sup35-NPD, respectively. These three Q-rich motifs exhibit a very strong positive relationship between STQ and STQN amino acid percentages and β-galactosidase activities (*Figure 4* and *Figure 5*). IUPred2A (https://iupred2a.elte.hu/plot_new), a web-server for identifying disordered protein regions (*Mészáros et al., 2018*), also revealed that Rad51-NTD, Rad53-SCD1 and Sup35-NPD are structurally flexible peptides. These results are consistent with the notion that, due to high STQ or STQN content, SCDs or Q-rich motifs are intrinsically disordered regions (IDRs) in their native states, rather than adopting stable secondary and/or tertiary structures (*Traven and Heierhorst, 2005*), and that a common feature of IDRs is their high content of S, T, Q, N, proline (P), glycine (G) and charged amino acids (*Romero et al., 2001*; *Macossay-Castillo et al., 2019*; *Uversky et al., 2000*).

It is important to note that the threshold of STQ or STQN content varies in the three cases presented herein (*Figure 4B*). Thus, the percentage of STQ or STQN residues is not likely the only factor contributing to protein expression levels. Since G, P, and glutamate (E) are enriched by >10% in Rad51-NTD, Rad53-SCD1, and Sup35-NPD, these three amino acids may also contribute to the PEE activities and structural flexibility of these three Q-rich motifs. Given that IDRs can endow proteins with structural and functional plasticity (*Zhou et al., 2019*; *Bondos et al., 2022*), we hypothesized

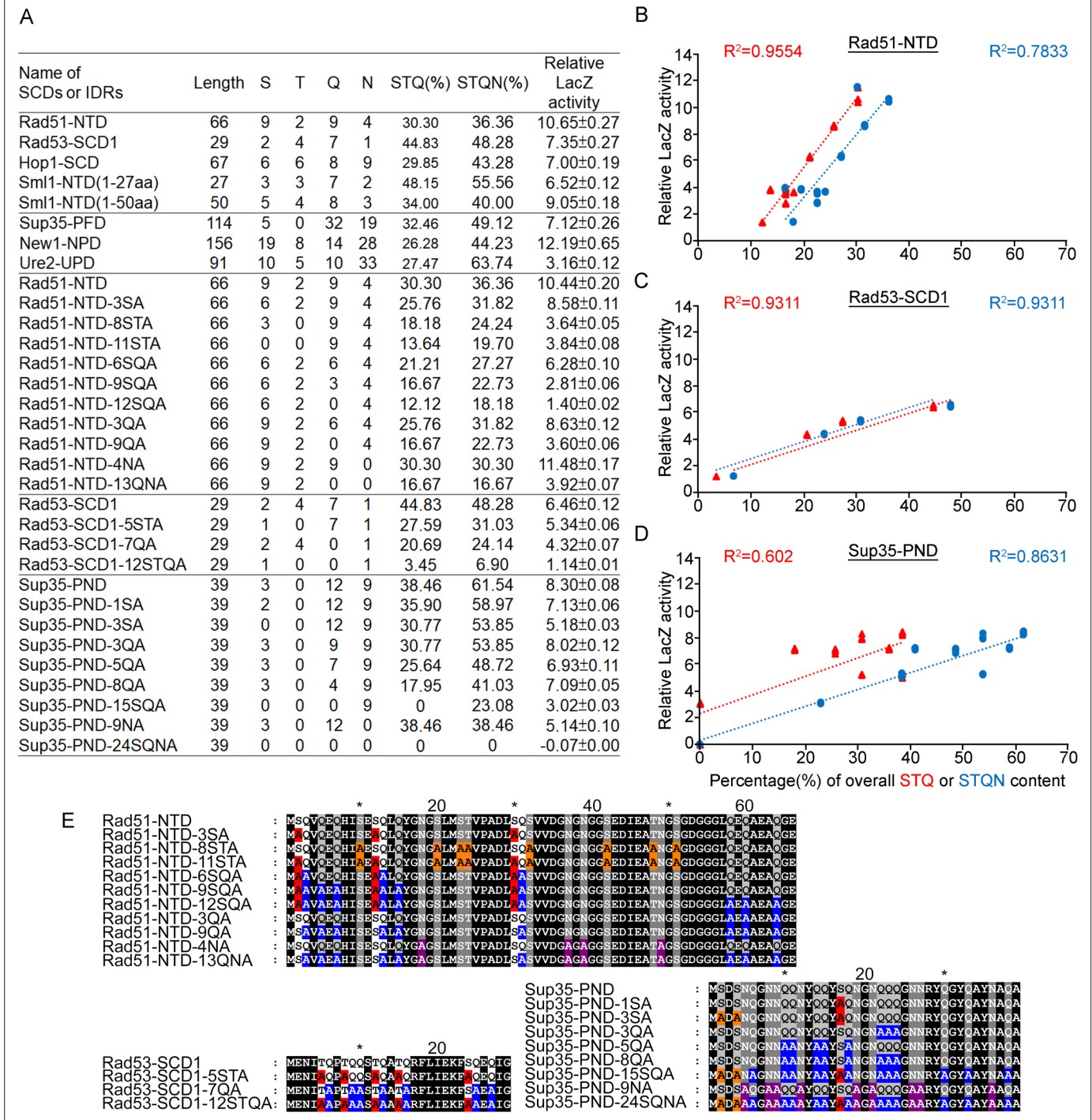

**Figure 4.** Relative β-galactosidase (LacZ) activities are correlated with the percentage STQ or STQN amino acid content of three Q-rich motifs. (**A**) List of N-terminal tags with their respective length, numbers of S/T/Q/N amino acids, overall STQ or STQN percentages, and relative β-galactosidase activities. (**B–D**) Linear regressions between relative β-galactosidase activities and overall STQ or STQN percentages for Rad51-NTD (**B**), Rad53-SCD1 (**C**) and Sup35-PND (**D**). The coefficients of determination ($R^2$) are indicated for each simple linear regression. (**E**) The amino acid sequences of wild-type and mutant Rad51-NTD, Rad51-SCD1 and Sup35-PND, respectively. Error bars are too small to be included.

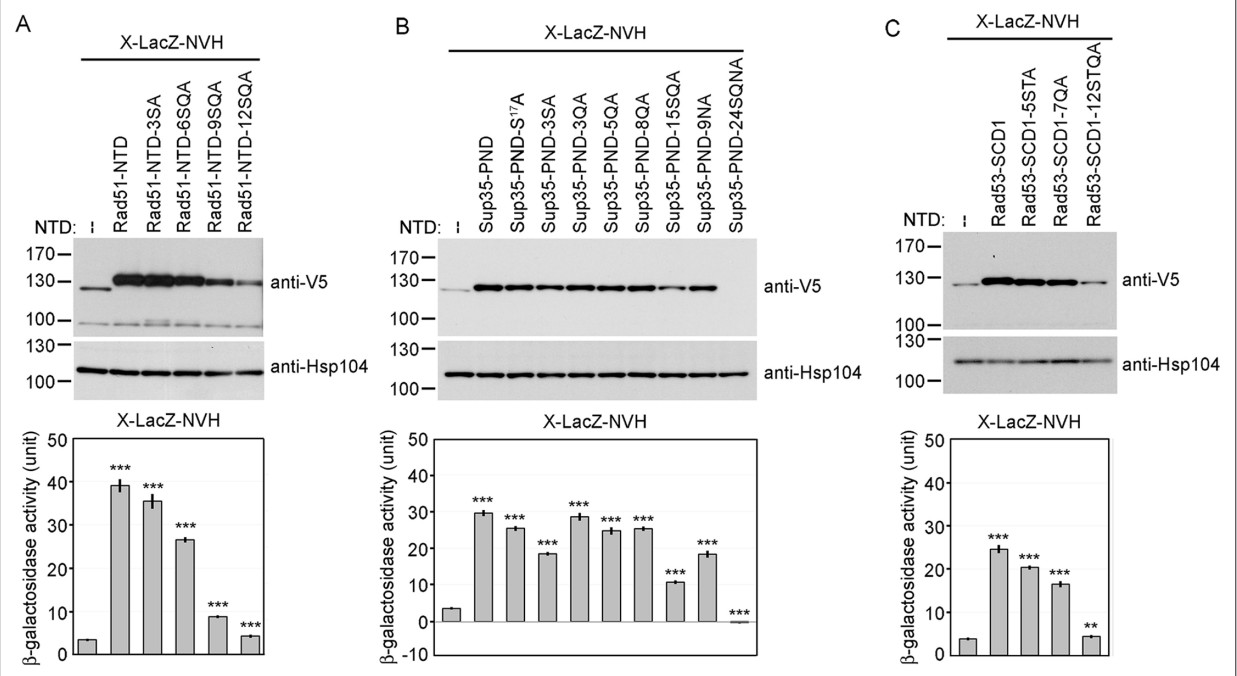

**Figure 5.** Alanine scanning mutagenesis of intrinsically disordered regions (IDRs). The amino acid sequences of WT and mutant IDRs are listed in *Supplementary file 1e*. Total protein lysates prepared from yeast cells expressing Rad51-NTD-LacZ-NVH (**A**), Sup35-PND-LacZ-NVH (**B**) or Rad53-SCD1-LacZ-NVH (**C**) were visualized by immunoblotting with anti-V5 antisera. Hsp104 was used as a loading control. Quantitative yeast β-galactosidase (LacZ) assays were carried out as described in *Figure 1*. Error bars indicate standard deviation between experiments (n=3). Asterisks indicate significant differences when compared to LacZ-NVH, with p values calculated using a two-tailed *t*-test (\*\*, p-value <0.01 and \*\*\*, p-value <0.001).

The online version of this article includes the following source data for figure 5:

**Source data 1.** Raw and labelled images for blots shown in *Figure 5*.

that Q-rich motifs (e.g. SCD, polyQ and polyQ/N) represent useful toolkits for creating new diversity during protein evolution.

## Comparative proteome-wide analyses of amino acid contents, SCDs and polyX motifs

Next, we designed five JavaScript software programs (AS-aa-content, AS-codon-usage, AS-Finder-SCD, AS-Finder-polyX and AS-Xcontent-7polyX) for proteome-wide analyses (*Supplementary file 1d*). AS-aa-content and AS-codon-usage determine the proteome-wide average contents of 20 different amino acids and the proteome-wide usage frequency of 64 genetic codons, respectively. ASFinder-SCD and ASFinder-polyX were applied to search for amino acid sequences that contain ≥3 S/T-Q motifs within a stretch of ≤100 residues (*Cheung et al., 2012*) and for the polyX motifs of 20 different amino acids, respectively. In the latter case, diverse thresholds have been used in different studies or databases to define and detect polyX motifs (*Mier and Andrade-Navarro, 2021*; *Ramazzotti et al., 2012*; *Li et al., 2016*; *Totzeck et al., 2017*). Based on a previous study (*Mier and Andrade-Navarro, 2021*), we applied seven different thresholds to seek both short and long, as well as pure and impure, polyX strings in 20 different representative near-complete proteomes, including 4 X (4/4), 5 X (4/5-5/5), 6 X (4/6-6/6), 7 X (4/7-7/7), 8–10 X (≥50% X), 11–10 X (≥50% X) and ≥21 X (≥50% X). The lowest threshold was ≥4/7, that is a minimum number of four identical X amino acid residues in a localized region of seven amino acid residues (*Figure 6*, *Figure 6—figure supplements 1–3*, and *Figure 6—source data 1–31*).

We then searched and compared the near-complete proteomes of 26 different eukaryotes (*Table 1*), including the budding yeast *S. cerevisiae*, three pathogenic species of *Candida*, three filamentous ascomycete fungi (*Neurospora crassa*, *Magnaporthe oryzae* and *Trichoderma reesei*), three basidiomycete fungi (*Cryptococcus neoformans*, *Ustilago maydis* and *Taiwanofungus camphoratus*), the slime mold *Dictyostelium discoideum*, the malaria-causing unicellular protozoan parasite *Plasmodium falciparum*,

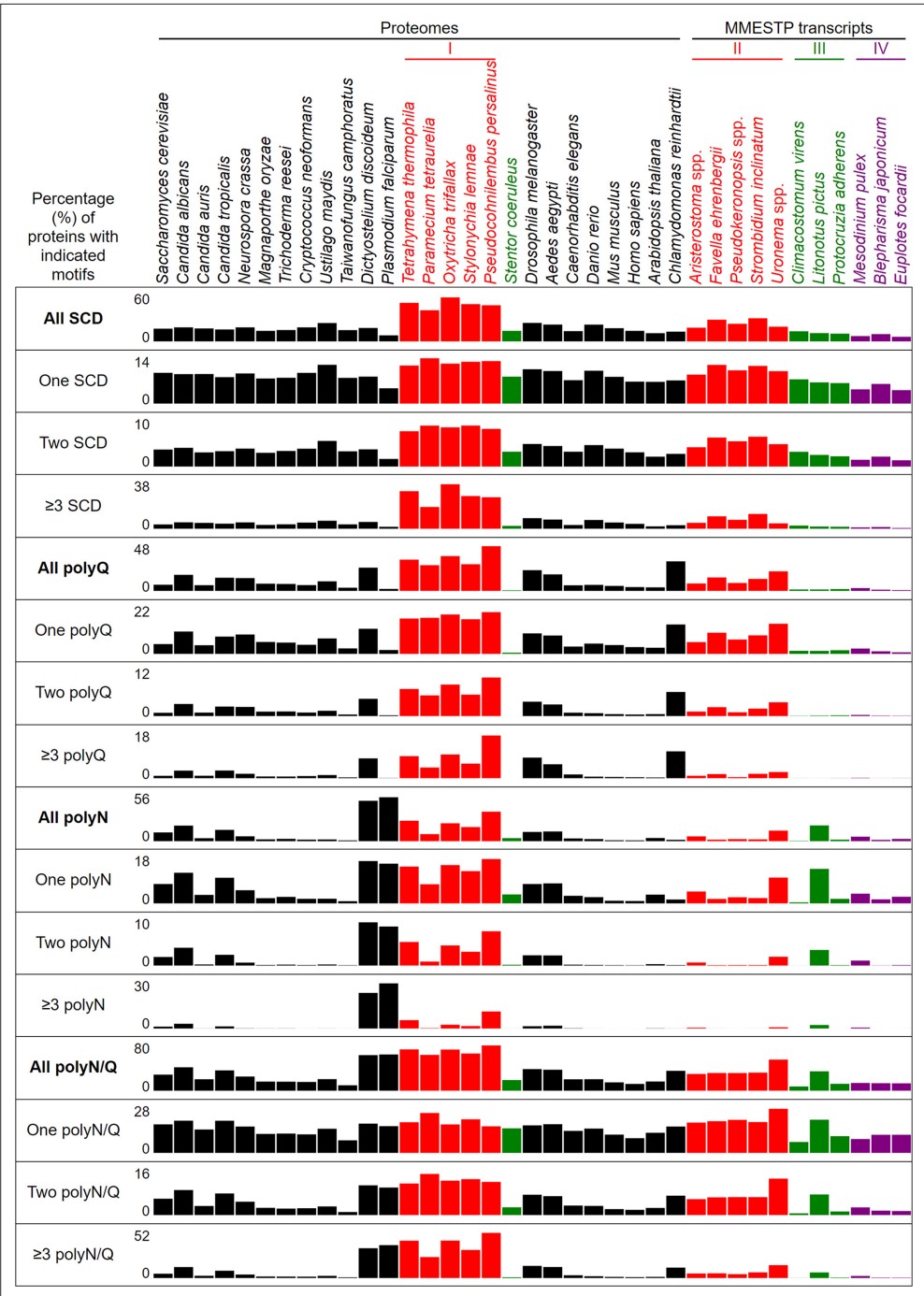

**Figure 6.** Percentages of proteins with different numbers of SCDs, and polyQ, polyQ/N or polyN tracts in 37 different eukaryotes.

The online version of this article includes the following source data and figure supplement(s) for figure 6:

**Source data 1.** The average usages of 20 different amino acids in 17 ciliate and 20 non-ciliate species.

**Source data 2.** The number of proteins containing different types of polyQ, polyQ/N and polyN tracts in 17 ciliate and 20 non-ciliate species.

**Source data 3.** The numbers and percentages of SCD and polyX proteins in 17 ciliate and 20 non-ciliate species.

**Source data 4.** The ratios of the overall number of X residues for each of the seven polyX motifs relative to those in the entire proteome of each species, respectively.

**Source data 5.** The codon usage frequency in 26 near-complete proteomes and 11 ciliate proteomes encoded by

*Figure 6 continued on next page*

*Figure 6 continued*

the transcripts generated as part of the Marine Microbial Eukaryote Transcriptome Sequencing Project (MMETSP).

**Source data 6.** GO enrichment analyses revealing the SCD and polyX proteins involved in different biological processes in 6 ciliate and 20 non-ciliate species.

**Source data 7.** GO enrichment analyses revealing the SCD and polyX proteins involved in different biological processes in 6 ciliate and 20 non-ciliate species.

**Source data 8.** GO enrichment analyses revealing the SCD and polyX proteins involved in different biological processes in 6 ciliate and 20 non-ciliate species.

**Source data 9.** GO enrichment analyses revealing the SCD and polyX proteins involved in different biological processes in 6 ciliate and 20 non-ciliate species.

**Source data 10.** GO enrichment analyses revealing the SCD and polyX proteins involved in different biological processes in 6 ciliate and 20 non-ciliate species.

**Source data 11.** GO enrichment analyses revealing the SCD and polyX proteins involved in different biological processes in 6 ciliate and 20 non-ciliate species.

**Source data 12.** GO enrichment analyses revealing the SCD and polyX proteins involved in different biological processes in 6 ciliate and 20 non-ciliate species.

**Source data 13.** GO enrichment analyses revealing the SCD and polyX proteins involved in different biological processes in 6 ciliate and 20 non-ciliate species.

**Source data 14.** GO enrichment analyses revealing the SCD and polyX proteins involved in different biological processes in 6 ciliate and 20 non-ciliate species.

**Source data 15.** GO enrichment analyses revealing the SCD and polyX proteins involved in different biological processes in 6 ciliate and 20 non-ciliate species.

**Source data 16.** GO enrichment analyses revealing the SCD and polyX proteins involved in different biological processes in 6 ciliate and 20 non-ciliate species.

**Source data 17.** GO enrichment analyses revealing the SCD and polyX proteins involved in different biological processes in 6 ciliate and 20 non-ciliate species.

**Source data 18.** GO enrichment analyses revealing the SCD and polyX proteins involved in different biological processes in 6 ciliate and 20 non-ciliate species.

**Source data 19.** GO enrichment analyses revealing the SCD and polyX proteins involved in different biological processes in 6 ciliate and 20 non-ciliate species.

**Source data 20.** GO enrichment analyses revealing the SCD and polyX proteins involved in different biological processes in 6 ciliate and 20 non-ciliate species.

**Source data 21.** GO enrichment analyses revealing the SCD and polyX proteins involved in different biological processes in 6 ciliate and 20 non-ciliate species.

**Source data 22.** GO enrichment analyses revealing the SCD and polyX proteins involved in different biological processes in 6 ciliate and 20 non-ciliate species.

**Source data 23.** GO enrichment analyses revealing the SCD and polyX proteins involved in different biological processes in 6 ciliate and 20 non-ciliate species.

**Source data 24.** GO enrichment analyses revealing the SCD and polyX proteins involved in different biological processes in 6 ciliate and 20 non-ciliate species.

**Source data 25.** GO enrichment analyses revealing the SCD and polyX proteins involved in different biological processes in 6 ciliate and 20 non-ciliate species.

**Source data 26.** GO enrichment analyses revealing the SCD and polyX proteins involved in different biological processes in 6 ciliate and 20 non-ciliate species.

**Source data 27.** GO enrichment analyses revealing the SCD and polyX proteins involved in different biological processes in 6 ciliate and 20 non-ciliate species.

**Source data 28.** GO enrichment analyses revealing the SCD and polyX proteins involved in different biological processes in 6 ciliate and 20 non-ciliate species.

**Source data 29.** GO enrichment analyses revealing the SCD and polyX proteins involved in different biological processes in 6 ciliate and 20 non-ciliate species.

**Source data 30.** GO enrichment analyses revealing the SCD and polyX proteins involved in different biological processes in 6 ciliate and 20 non-ciliate species.

*Figure 6 continued*

**Source data 31.** GO enrichment analyses revealing the SCD and polyX proteins involved in different biological processes in 6 ciliate and 20 non-ciliate species.

**Source data 32.** The results of BLASTP searches using the 58 *Tetrahymena thermophila* proteins involved in xylan catabolysis.

**Source data 33.** The list of 124 *Tetrahymena thermophila* proteins involved in meiosis (kindly provided by Josef Loidl).

**Figure supplement 1.** Proteome-wide contents of 20 different amino acids in 37 different eukaryotes.

**Figure supplement 2.** Percentages of proteins with indicated polyQ and polyQ/N tracts in 37 different eukaryotes.

**Figure supplement 3.** Percentages of proteins with indicated polyX motifs in 37 different eukaryotes.

---

six unicellular ciliates (*Tetrahymena thermophila, Paramecium tetraurelia, Oxytricha trifallax, Stylonychia lemnae, Pseudocohnilembus persalinus* and *Stentor coeruleus*), the fly *Drosophila melanogaster*, the mosquito *Aedes aegypti,* the nematode *Caenorhabditis elegans*, the zebrafish *Danio rerio*, the mouse *Mus musculus*, *Homo sapiens*, the higher plant *Arabidopsis thaliana*, and the single-celled green alga *Chlamydomonas reinhardtii*. The Benchmarking Universal Single-Copy Ortholog (BUSCO) scores of the near-universal single-copy gene orthologs of all 27 proteomes are 92.4–100% (*Table 1*). Genome or protein matrix scores >95% for model organisms are generally deemed complete reference genomes or proteomes (*Seppey et al., 2019*).

It was reported previously that SCDs are overrepresented in the yeast and human proteomes (*Cheung et al., 2012*; *Cara et al., 2016*), and that polyX prevalence differs among species (*Mier et al., 2020*; *Kuspa and Loomis, 2006*; *Davies et al., 2017*; *Mier et al., 2017*). Our results reveal that the percentages of SCD proteins in the near-complete proteomes of 21 non-ciliate species and 6 ciliates range from 8.0% in *P. falciparum*, 13.9% in *H. sapiens*, 16.8% in *S. cerevisiae*, 24.2% in *U. maydis*, to a maximum of 58.0% in *O. trifallax* (*Figure 6* and *Figure 6—source data 2*). Among the 6050 proteins in the most recently updated *S. cerevisiae* reference proteome (https://www.uniprot.org/proteomes/UP000002311), we identified 1016 SCD-hosting proteins (*Figure 6—source data 2*), including all 436 SCD-harboring proteins previously revealed by ScanProsite (*Cheung et al., 2012*). ScanProsite is a publicly available database of protein families, domains and motifs (*de Castro et al., 2006*).

The most striking finding in our study is that, due to their usage of the two noncanonical codons (UAA[Q] and UAG[Q]), Q (but not S, T or N) is used more frequently in five unicellular ciliates (i.e. *T. thermophila*, *P. tetraurelia*, *O. trifallax*, *S. lemnae,* and *P. persalinus*) than in eukaryotes with standard genetic codons, including the unicellular ciliate *S. coeruleus* and all of the 20 non-ciliate species we examined herein (*Figure 6—figure supplement 1* and *Figure 6—source data 1*). Hereafter, we refer to the five unicellular ciliates with reassigned stop codons as 'group I' ciliates. Due to higher proteome-wide Q contents, there are higher percentages of SCD, polyQ, and polyQ/N in the five group I ciliates than in *S. coeruleus* (*Figure 6*, *Figure 6—figure supplements 2–3*, and *Figure 6—source data 1–3*).

Next, we analyzed the SCD and polyX proteins encoded by the transcriptomes of 11 different ciliate species. These transcripts were originally generated as part of the Marine Microbial Eukaryote Transcriptome Sequencing Project (MMETSP) (*Keeling et al., 2014*), which were then reassembled and reannotated by Brown and colleagues (*Johnson et al., 2019*). All transcripts are publicly available from Zendo (https://zenodo.org/record/1212585#.Y79zoi2l3PA; *Johnson et al., 2019*). We applied TransDecoder (https://github.com/TransDecoder/TransDecoder/wiki; *Haas, 2023*) to identify candidate coding regions within the transcript sequences. Five of those 11 ciliates have reassigned UAA[Q] and UAG[Q] codons (hereafter termed 'group II ciliates'), that is *Aristerostoma* spp., *Favella ehrenbergii*, *Pseudokeronopsis* spp., *Strombidium inclinatum,* and *Uronema* spp. Like sessile *S. coeruleus*, group III ciliates (*Climacostomum virens, Litonotus pictus* and *Protocruzia adherens*) possess the standard genetic codes. Group IV ciliates encompass *Mesodinium pulex*, *Blepharisma japonicum* and *Euplotes focardii*, each of which utilizes the reassigned codons UAA[Y], UAG[Y], UGA[W], and UGA[C], respectively (*Table 1*). For two reasons, the proteins encoded by these MMESTP transcripts are unlikely to represent the entire protein complement of all 11 ciliate species. First, many MMETSP transcripts are not intact (i.e., broken mRNAs) and thus encode incomplete protein sequences. Second, except for *C. virens* (94.7%), the BUSCO protein scores of these MMESTP transcripts only range from 52.6% to

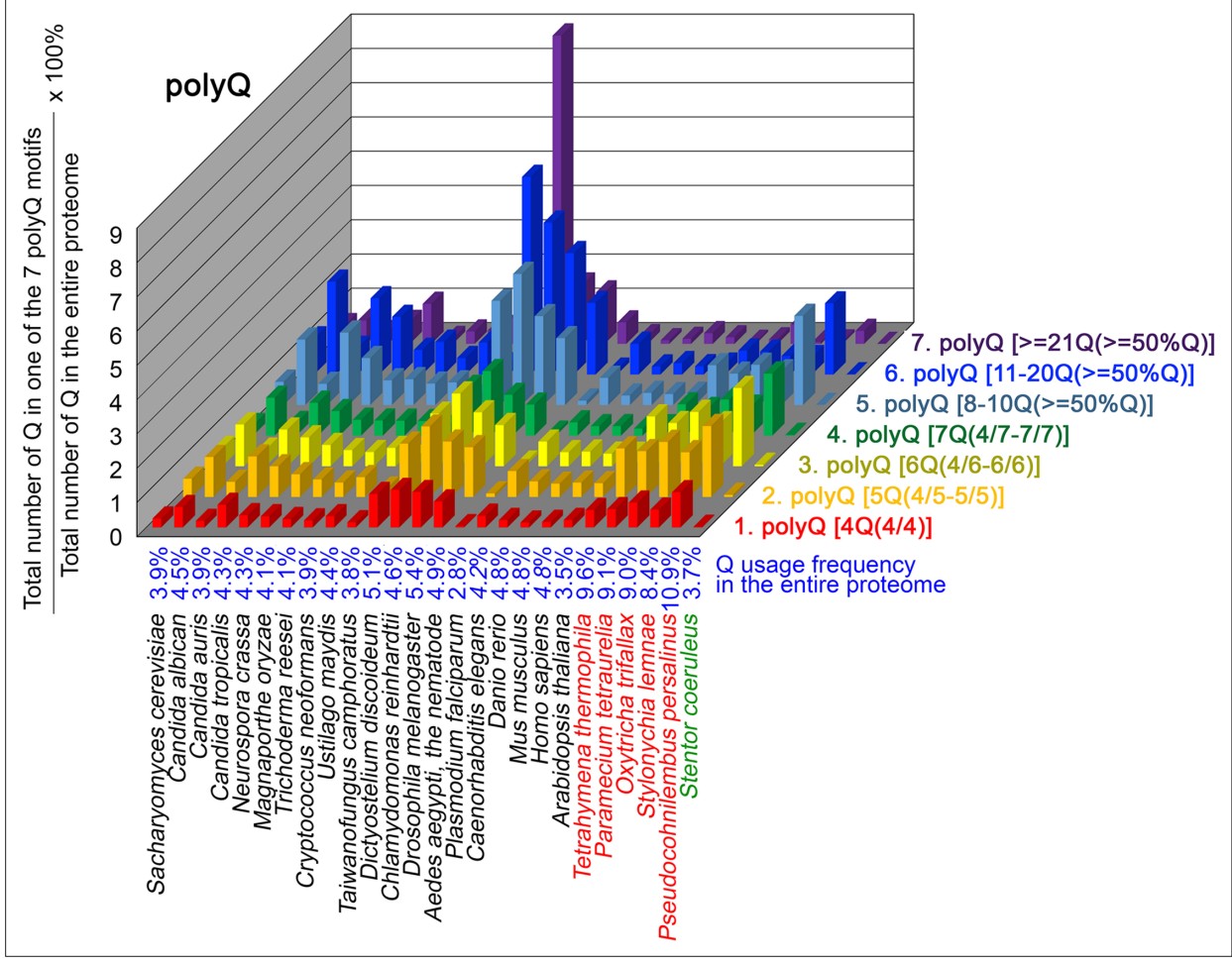

**Figure 7.** Q contents in 7 different types of polyQ motifs in 26 near-complete proteomes. The five ciliates with reassigned stops codon (TAA$^Q$ and TAG$^Q$) are indicated in red. *Stentor coeruleus,* a ciliate with standard stop codons, is indicated in green.

89.9% (*Table 1*). Nevertheless, our results indicate that Q is used more frequently in group I and group II ciliates than in group III and group IV ciliates or in a further 20 non-ciliate species (*Figure 6—figure supplement 1*, and *Figure 6—source data 1*). Accordingly, proportions of SCD, polyQ and polyQ/N proteins in all group I and group II ciliates are higher than they are in the three group III ciliates (except *L. pictus*) and the three group IV ciliates, respectively. Since N is used more frequently in *L. pictus* than the other ciliates in groups II-IV, it has higher percentages of polyN and polyQ/N proteins (*Figure 6*, *Figure 6—figure supplements 2 and 3*, and *Figure 6—source data 1–3*). Our data also indicates that Y, W, or C are not used more frequently in the three group IV ciliates than in the other 14 ciliate or 20 non-ciliate species (*Figure 6—figure supplement 1*, and *Figure 6—source data 1* file 1). Reassignments of stop codons to Y, W, or C also do not result in higher percentages of polyY, polyW, or polyC proteins in the three group IV ciliates, respectively (*Figure 6—figure supplement 3*).

To further confirm the above-described results, we normalized the runs of amino acids and created a null expectation from each proteome by determining the ratios of the overall number of X residues for each of the seven polyX motifs relative to those in the entire proteome of each species, respectively. The results for four different polyX motifs, that is polyQ, polyN, polyS and polyT, are presented in *Figures 7–10* and *Figure 6—source data 4*. The results summarized in *Figures 7–10* support that polyX prevalence differs among species and that the overall X contents of polyX motifs often but not always correlate with the X usage frequencies in entire proteomes (*Mier et al., 2020*). Most importantly, our results reveal that, compared to *S. coeruleus* or several non-ciliate eukaryotic organisms (e.g. *P. falciparum, C. elegans, D. rerio, M. musculus,* and *H. sapiens*), the five ciliates with reassigned TAA$^Q$ and TAG$^Q$ codons not only have higher Q usage frequency but also more polyQ

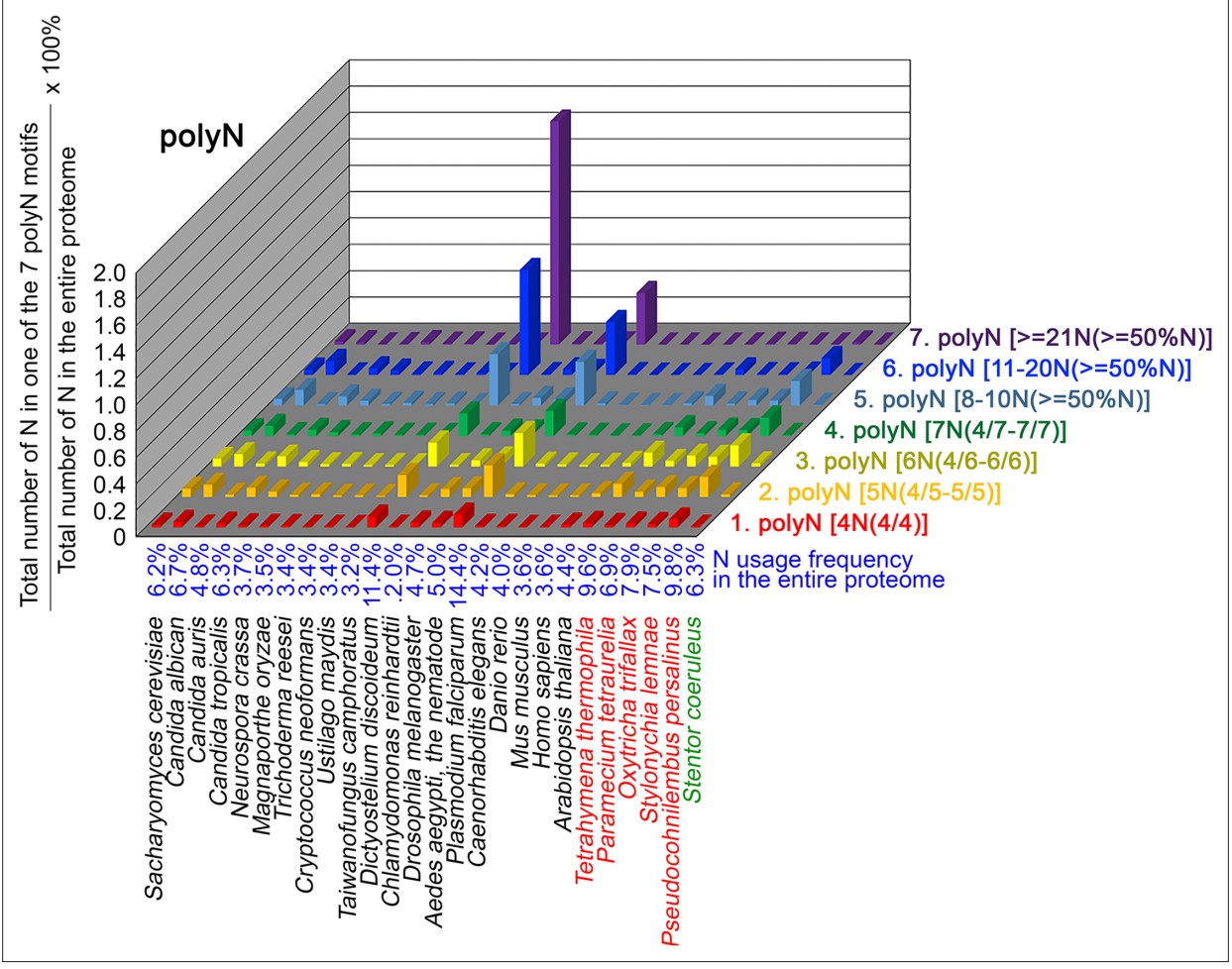

**Figure 8.** N contents in 7 different types of polyN motifs in 26 near-complete proteomes. The five ciliates with reassigned stops codon (TAA$^Q$ and TAG$^Q$) are indicated in red. *Stentor coeruleus,* a ciliate with standard stop codons, is indicated in green.

motifs in their proteomes (*Figure 7* and *Figure 6—source data 4*). In contrast, polyQ motifs prevail in *C. albicans, C. tropicalis, D. discoideum, C. reinhardtii, D. melanogaster,* and *A. aegypti,* although the Q usage frequencies in their entire proteomes are not significantly higher than those of other eukaryotes (*Figure 7* and *Figure 6—source data 4*). Due to their higher N usage frequencies, *D. discoideum, P. falciparum,* and *P. persalinus* have more polyN motifs than the other 23 eukaryotes we examined here (*Figure 8* and *Figure 6—source data 4*). Generally speaking, all 26 eukaryotes we assessed have similar S usage frequencies and percentages of S contents in polyS motifs (*Figure 9* and *Figure 6—source data 4*). Among these 26 eukaryotes, *D. discoideum* possesses many more polyT motifs, although its T usage frequency is similar to that of the other 25 eukaryotes (*Figure 10* and *Figure 6—source data 4*). Several other polyX motifs are particularly enriched in specific eukaryotes, for example, polyK and polyY in *P. falciparum,* polyK and polyF in *D. discoideum,* polyG, polyA, polyP, and polyW in *C. reinhardtii,* as well as the longest polyC (i.e. ≥21 C and ≥50% C) in *C. tropicalis* (*Figure 6—source data 4*). Further investigations will decipher the structural and functional relevance of those polyX motif proteins. In conclusion, these normalized results further confirm that reassignment of stop codons to Q indeed results in both higher Q usage frequencies and more polyQ motifs in ciliates.

## The frequency of TAA$^Q$ and TAG$^Q$, CAA$^Q$ and CAG$^Q$ usage in 26 different organisms

PolyQ motifs have a particular length-dependent codon usage that relates to strand slippage in CAG/CTG trinucleotide repeat regions during DNA replication (*Petruska et al., 1998*; *Mier and Andrade-Navarro, 2021*). In most organisms having standard genetic codons, Q is encoded by CAG$^Q$ and

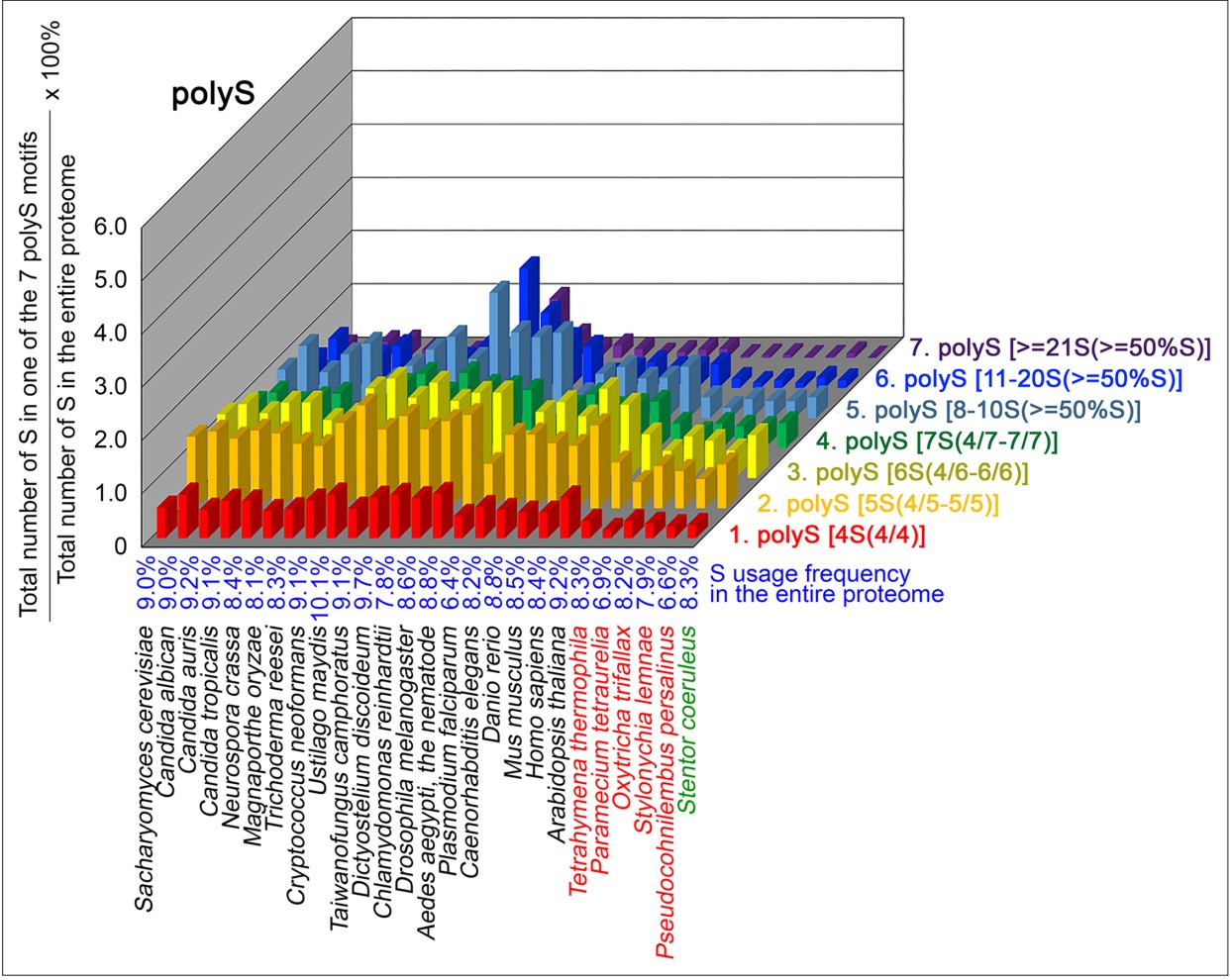

**Figure 9.** contents in 7 different types of polyS motifs in 26 near-complete proteomes. The five ciliates with reassigned stops codon (TAA$^Q$ and TAG$^Q$) are indicated in red. *Stentor coeruleus,* a ciliate with standard stop codons, is indicated in green.

CAA$^Q$. We applied AS-Xcontent, a JavaScript software program (*Supplementary file 1d*), to determine and compare proteome-wide Q contents, as well as CAG$^Q$ usage frequencies (i.e. the ratio between CAG$^Q$ and the sum of CAG$^Q$, CAG$^Q$, TAA$^Q$, and TAG$^Q$) (*Table 2* and *Figure 6—source data 5*). Our results reveal that the likelihood of forming long CAG/CTG trinucleotide repeats is higher in five eukaryotes due to their higher CAG$^Q$ usage frequencies, including *D. melanogaster* (86.6% Q), *D. rerio* (74.0% Q), *M. musculus* (74.0% Q), *H. sapiens* (73.5% Q), and *C. reinhardtii* (87.3% Q) (orange background, *Table 2*). In contrast, another five eukaryotes that possess high numbers of polyQ motifs (i.e. *D. discoideum, C. albicans, C. tropicalis, P. falciparum* and *S. coeruleus*) (*Figure 7*) utilize more CAA$^Q$ (96.2%, 84.6%, 84.5%, 86.7%, and 75.7%) than CAG$^Q$ (3.8%, 15.4%, 15.5%, 13.3%, and 24.3%), respectively, to avoid forming long CAG/CTG trinucleotide repeats (green background, *Table 2*). Similarly, all five ciliates with reassigned stop codons (TAA$^Q$ and TAG$^Q$) display low CAG$^Q$ usage frequencies (i.e. ranging from 3.8% Q in *P. persalinus* to 12.6% Q in *O. trifallax*) (*Table 2*). Accordingly, the CAG-slippage mechanism may operate more frequently in *C. reinhardtii, D. melanogaster, D. rerio, M. musculus* and *H. sapiens* than in *D. discoideum, C. albicans, C. tropicalis, P. falciparum, S. coeruleus* and the five ciliates with reassigned stop codons (TAA$^Q$ and TAG$^Q$).

## Q-rich-motif proteins are overrepresented in specialized biological processes of various eukaryotic proteomes

To determine the biological impacts of Q-rich-motif proteins, we designed a JavaScript software tool AS-GOfuncR-FWER (*Supplementary file 1d*) to carry out comparative Gene Ontology (GO)

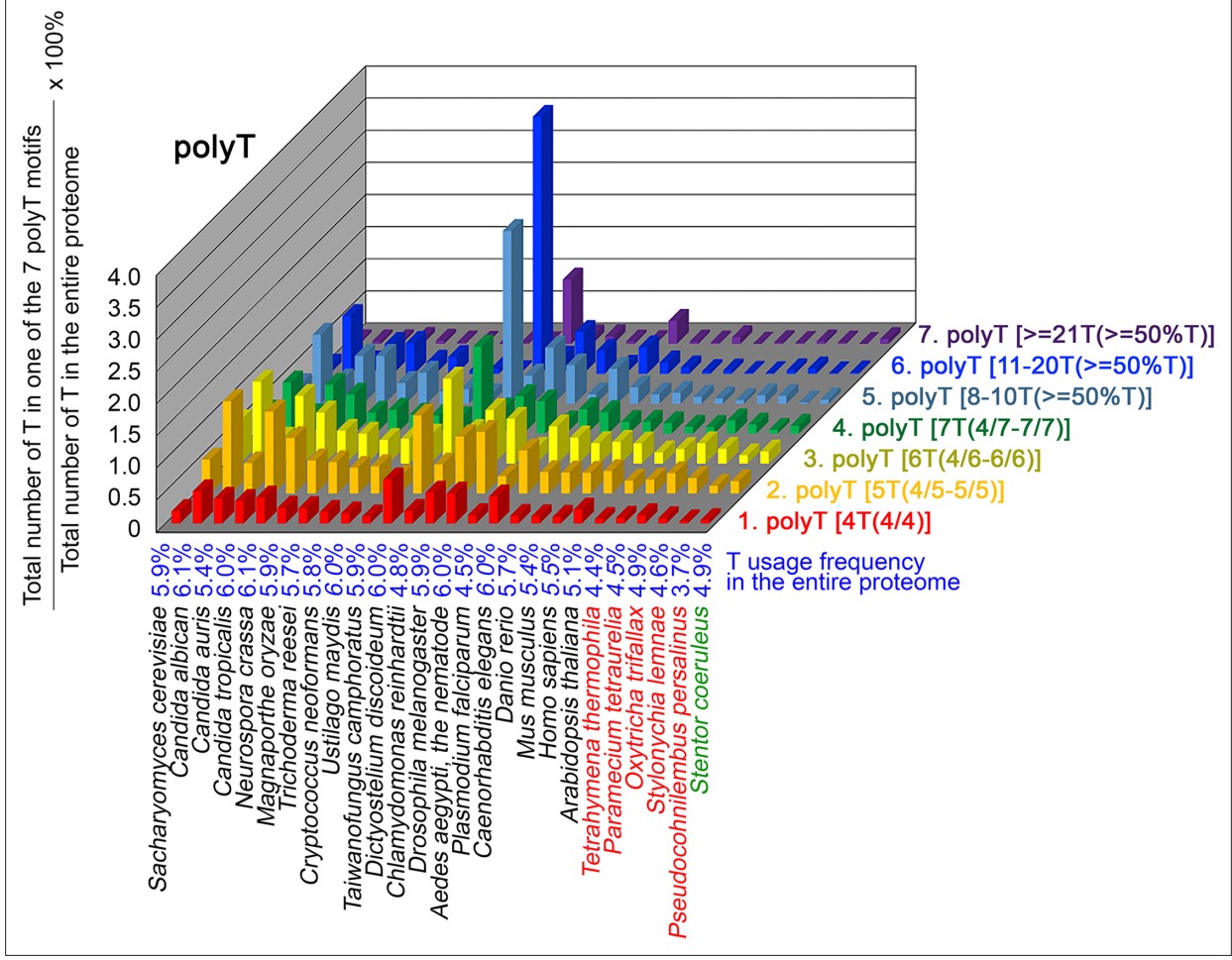

**Figure 10.** T contents in 7 different types of polyT motifs in 26 near-complete proteomes. The five ciliates with reassigned stops codon (TAA^Q and TAG^Q) are indicated in red. *Stentor coeruleus,* a ciliate with standard stop codons, is indicated in green.

enrichment analyses using information on the functions of genes provided by the GO knowledge-base (http://geneontology.org). Rigorous statistical testing for overrepresentation or underrepresentation of SCD and polyX proteins was performed using GOfuncR (https://bioconductor.org/packages/release/bioc/html/GOfuncR.html), an R package program that conducts standard candidate vs. background enrichment analysis employing the hypergeometric test. The raw p-values were adjusted according to the Family-Wise Error Rate (FWER). The same method was applied to the GO enrichment analysis of human genomes (*Huttenhower et al., 2009*). The results presented in *Figure 11* and *Figure 12*, *Figure 6—source data 1–31* support the hypothesis that Q-rich motifs prevail in proteins involved in specialized biological processes, including *S. cerevisiae* RNA-mediated transposition, *C. albicans* filamentous growth, peptidyl-glutamic acid modification in ciliates with reassigned stop codons (TAA^Q and TAG^Q), *T. thermophila* xylan catabolism, *D. discoideum* sexual reproduction, *P. falciparum* infection, as well as the nervous systems of *D. melanogaster, M. musculus,* and *H. sapiens*. In contrast, peptidyl-glutamic acid modification is not overrepresented with Q-rich-motif proteins in *S. coeruleus,* a ciliate with standard stop codons.

Our results are also consistent with a previous report that there is an overrepresentation of conserved Q-rich-motif proteins in pathways related to the human nervous system (*Cara et al., 2016*). For instance, human CTTNBP2 (1663 amino acid residues) is a neuron-specific F-actin-associated SCD protein that is involved in the formation and maintenance of dendritic spines and it is associated with autism spectrum disorders (*Chen and Hsueh, 2012*; *Hsueh, 2012*). Human CTTNBP2 possesses ten S/T-Q motifs ($T^{466}Q$, $T^{493}Q$, $S^{1580}Q$, $S^{553}Q$, $S^{634}Q$, $S^{994}Q$, $S^{1392}Q$, $S^{1580}Q$, $T^{1621}Q$ and $S^{1624}Q$). Mouse CTTNBP2 has 630 amino acid residues, four S/T-Q motifs ($S^{419}Q$, $T^{463}Q$, $S^{550}Q$, $S^{624}Q$), and it shares

**Table 2.** Usage frequencies of TAA[*], TAG[*], TAA[Q], TAG[Q], CAA[Q], and CAG[Q] codons in the entire proteomes of 26 different organisms.

| Species | CAA | CAG | TAA | TAG |
|---|---|---|---|---|
| *Saccharomyces cerevisiae S288c* | 2.73 (62.6%Q) | 1.21 (37.4%Q) | 0.11 | 0.05 |
| *Candida albicans* | 3.57 (84.6%Q) | 0.65 (15.4%Q) | 0.1 | 0.05 |
| *Candida auris* | 1.81 (46.1%Q) | 2.12 (53.9%Q) | 0.08 | 0.06 |
| *Candida tropicalis* | 3.61 (84.5%Q) | 0.66 (15.5%Q) | 0.1 | 0.07 |
| *Neurospora crassa* | 1.70 (39.5%Q) | 2.60 (60.5%Q) | 0.06 | 0.05 |
| *Magnaporthe oryzae* | 1.37 (33.7%Q) | 2.69 (66.3%Q) | 0.06 | 0.07 |
| *Trichoderma reesei* | 1.17 (28.4%Q) | 2.95 (71.6%Q) | 0.06 | 0.06 |
| *Cryptococcus neoformans* | 2.06 (53.5%Q) | 1.79 (46.5%Q) | 0.07 | 0.06 |
| *Ustilago maydis* | 1.82 (41.3%Q) | 2.61 (58.7%Q) | 0.04 | 0.05 |
| *Taiwanofungus camphoratus* | 1.57 (41.8%Q) | 2.19 (58.2%Q) | 0.05 | 0.06 |
| *Dictyostelium discoideum* | 4.86 (96.2%Q) | 0.19 (3.8%Q) | 0.16 | 0.01 |
| *Plasmodium falciparum* | 2.42 (86.7%Q) | 0.37 (13.3%Q) | 0.09 | 0.01 |
| *Drosophila melanogaster* | 1.56 (13.4%Q) | 3.61 (86.6%Q) | 0.08 | 0.07 |
| *Aedes aegypti* | 1.76 (40.6%Q) | 2.58 (59.4%Q) | 0.11 | 0.07 |
| *Caenorhabditis elegans* | 2.74 (65.6%Q) | 1.44 (34.4%Q) | 0.16 | 0.06 |
| *Danio rerio* | 1.18 (26.0%Q) | 3.35 (74.0%Q) | 0.11 | 0.06 |
| *Mus musculus* | 1.20 (26.0%Q) | 3.41 (74.0%Q) | 0.1 | 0.08 |
| *Homo sapiens* | 1.23 (26.5%Q) | 3.42 (73.5%Q) | 0.1 | 0.08 |
| *Arabidopsis thaliana* | 1.94 (56.1%Q) | 1.52 (43.9%Q) | 0.09 | 0.05 |
| *Chlamydomonas reinhardtii* | 0.59 (12.7%Q) | 4.05 (87.3%Q) | 0.03 | 0.04 |
| *Tetrahymena thermophila* | 2.04 (21.2%Q) | 0.48 (5.0%Q) | 5.46 (56.8%Q) | 1.63 (17.0%Q) |
| *Paramecium tetraurelia* | 2.54 (27.9%Q) | 0.57 (6.3%Q) | 4.53 (46.7%Q) | 1.48 (16.2%Q) |
| *Oxytricha trifallax* | 2.68 (29.9%Q) | 1.07 (12.0%Q) | 3.63 (40.6%Q) | 1.57 (17.5%Q) |
| *Stylonychia lemnae* | 2.26 (21.1%Q) | 1.05 (12.6%Q) | 3.22 (38.6%Q) | 1.81 (21.7%Q) |
| *Pseudocohnilembus persalinus* | 1.76 (18.0%Q) | 0.37 (3.8%Q) | 7.36 (76.0%Q) | 1.39 (14.4%Q) |
| *Stentor coeruleus* | 2.77 (75.7%Q) | 0.89 (24.3%Q) | 0.16 | 0.08 |

a high amino acid identity with the N-terminus (1–640 amino acid residues) of human CTTNBP2. IUPred2A (https://iupred2a.elte.hu/plot_new) also reveals that both human CTTNBP2 (220–1633 residues) and mouse CTTNBP2 (220–630 residues) are Q/N-rich IDRs with high percentages of S, T, Q, N, G, P, R, and K. We reported recently that mouse CTTNBP2 forms self-assembled condensates through its C-terminal IDR and it facilitates co-condensation of an abundant excitatory postsynaptic scaffold protein SHANK3 at dendritic spines in a $Zn^{2+}$-dependent manner (*Shih et al., 2022*).

## Q-rich-motif proteins prevail in the *T. thermophila* xylan catabolic process

The proteome of *T. thermophila* contains 58 proteins involved in the xylan catabolic process (GO ID: 45493), of which 56 (97%), 55 (95%), 58 (100%), and 49 (84%) proteins harbor SCD, polyQ, polyQ/N, and polyN tracts, respectively (*Figure 6—source data 18–23*). Using the NCBI BLASTP search tool with an expect value (E-value) ≤10e-5 to search for homologs of these 58 proteins among all the other 16 ciliates analyzed in this study, we only identified 144 proteins with amino acid identity >60% and

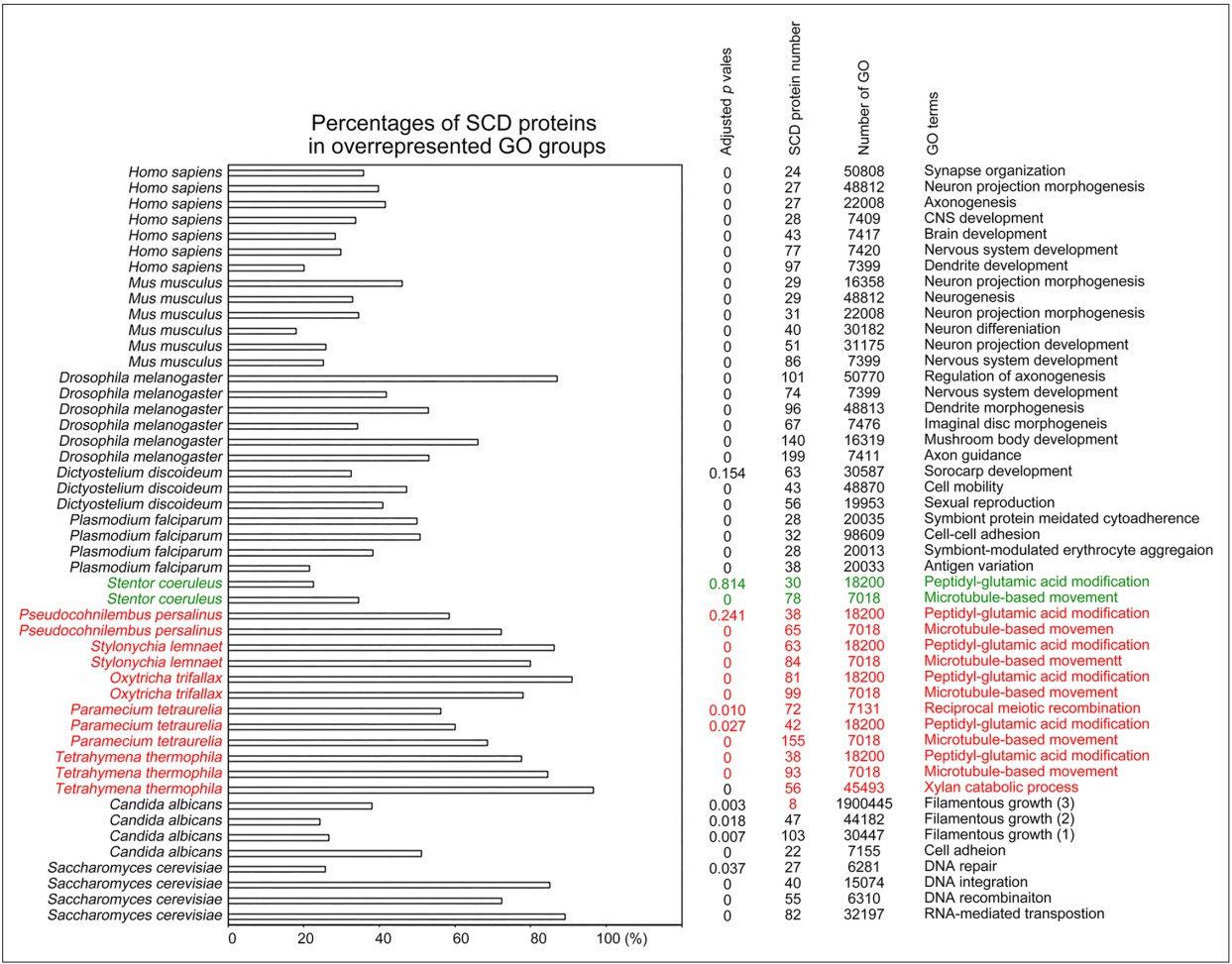

**Figure 11.** Selection of biological processes with overrepresented SCD-containing proteins in different eukaryotes. The percentages and number of SCD-containing proteins in our search that belong to each indicated Gene Ontology (GO) group are shown. GOfuncR (*Huttenhower et al., 2009*) was applied for GO enrichment and statistical analysis. The p values adjusted according to the Family-wise error rate (FWER) are shown.

a raw alignment score of >150 (*Figure 6—source data 32*). Thus, *T. thermophila* has more abundant xylan catabolic proteins than all other ciliates we examined herein.

## Most proteins involved in *T. thermophila* meiosis harbor one or more Q-rich motif(s)

Ciliate meiosis is remarkable relative to that of other studied sexual eukaryotes. Ciliates often have two types of nuclei. Their diploid micronucleus (MIC) carries the cell germline, the genetic material of which is inherited via sexual reproduction and meiosis. The polyploid macronucleus (MAC) or vegetative nucleus provides nuclear RNA for vegetative growth. The MAC is generated from the MIC by massive amplification, editing and rearrangement of the genome (see reviews in *Prescott, 1994*; *Chalker and Yao, 2011*). In *T. thermophila*, the most intensively studied ciliate, meiotic MICs undergo extreme elongation (by ~50-fold) and form proteinaceous condensates called 'crescents'. Within these elongated crescents, telomeres and centromeres of all meiotic chromosomes are rearranged at opposing ends in a stretched bouquet-like manner. Meiotic pairing and recombination take place within the crescents (see review in *Loidl, 2021*). It has been reported that ATR1 (Ataxia Telangiectasia Mutated 1), an evolutionarily conserved DNA damage senor protein kinase, senses Spo11-induced DSBs and triggers the elongation of MICs (*Loidl and Mochizuki, 2009*). Meiosis-specific CYC2 and CYC17 cyclins, as well as cyclin-dependent kinase CDK3, are required to initiate meiosis and for crescent assembly (*Yan et al., 2016a*; *Yan et al., 2016b*; *Xu et al., 2019*). CYC2/CDK2 promotes bouquet formation in MICs by controlling microtubule-directed elongation (*Xu et al., 2019*) and it

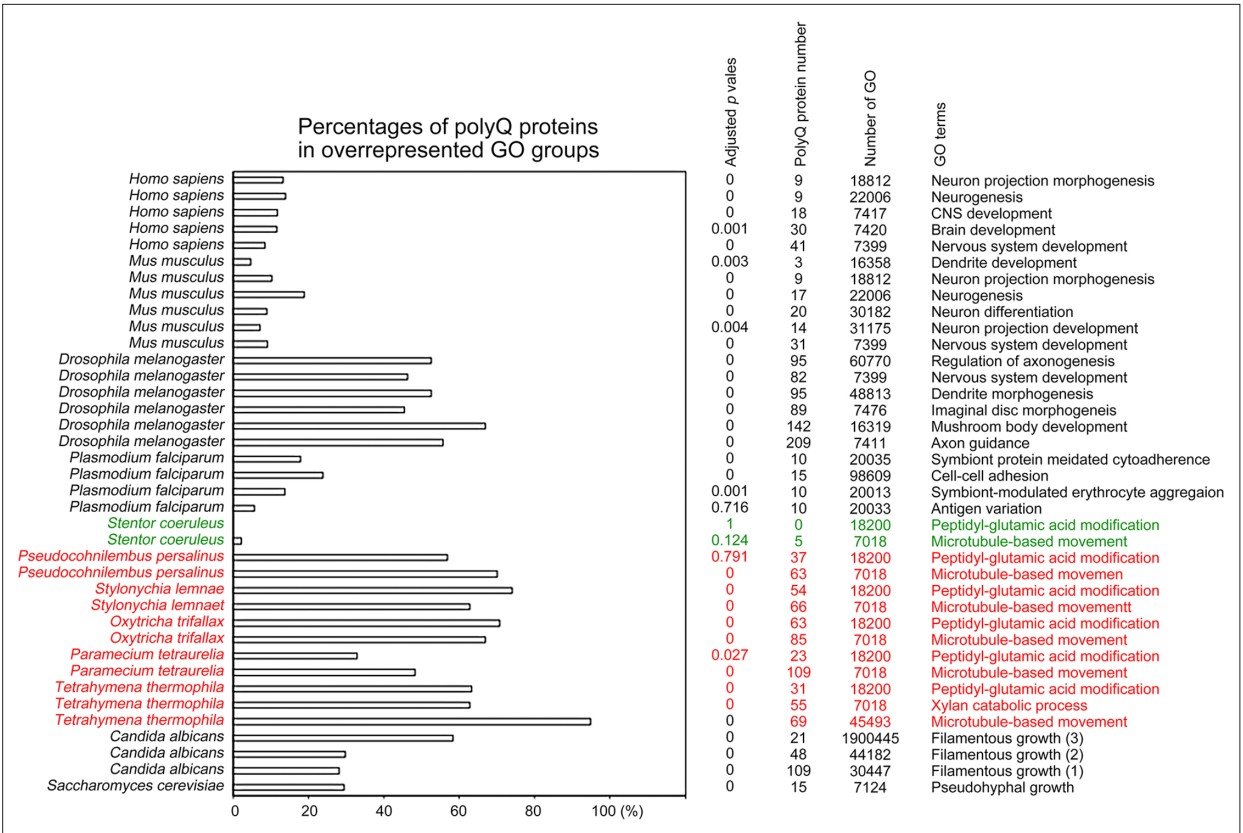

**Figure 12.** Selection of biological processes with overrepresented polyQ-containing proteins in different eukaryotes. The percentages and numbers of polyQ-containing proteins in our search that belong to each indicated Gene Ontology (GO) group are shown. GOfuncR (*Huttenhower et al., 2009*) was applied for GO enrichment and statistical analysis. The *p* values adjusted according to the Family-wise error rate (FWER) are shown. The five ciliates with reassigned stops codons (TAA$^Q$ and TAG$^Q$) are indicated in red. *Stentor coeruleus,* a ciliate with standard stop codons, is indicated in green.

also controls the gene expression of proteins involved in DSB formation (SPO11), DNA repair (COM1, EXO1, DMC1), and crossover formation (HOP2, MND1, MSH4, MSH5, ZPH3, BIM1, and BIM2) (*Zhang et al., 2018*). The DPL2/E2fl1 complex, a meiosis-specific transcription factor, promotes transcriptional induction of DNA repair proteins and chromosome-associated structural proteins, including MRE11, COM1, EXO1, RAD50, RAD51, SMC1, SMC2, SMC3, SMC4, REC8, ESP1, and CNA1, among others (*Zhang et al., 2018*). Nevertheless, the molecular mechanisms underlying crescent assembly and disassembly remain poorly understood.

Among the 124 *T. thermophila* meiotic proteins (*Figure 6—source data 33*, see review in *Loidl, 2021*), we identified 85 SCD proteins, 54 polyQ proteins, 106 polyQ/N proteins, 32 polyN proteins, 32 polyS proteins, and 48 polyK proteins, respectively. Notably, there are 48 and 59 meiotic proteins that contain ≥4 SCDs and/or ≥4 polyQ/N tracts, respectively. For instance, DPL2, CYC2, CYC17, and ATR1 each contain 15, 6, 8 and 5 SCDs, 2, 0, 1 and 2 polyQ tracts, as well as 11, 4, 4 and 5 polyQ/N tracts, respectively. Pars11, a chromatin-associated protein required for inducing and limiting Spo11-induced DSBs, has 14 SCDs, 2 polyQ tracts and 6 polyQ/N tracts. Spo11-induced DSBs promote ATR1-dependent Pars11 phosphorylation and its removal from chromatin (*Zhang et al., 2018*). Many *T. thermophila* meiotic DSB repair proteins also harbor several SCDs, polyQ tracts and/or polyQ/N tracts, including MSH4, MSH5, SGS1, FANCM, REC8, and ZPH3, among others.

Several *T. thermophila* proteins involved in editing and rearrangement of the MIC genome also harbor multiple Q-rich motifs. PDD1 (programmed DNA degradation 1), a conjugation-specific HP1-like protein, has 13 SCDs, 3 polyQ tracts and 6 polyQ/N tracts. Mutations in the chromodomain or the chromoshadow domain of PDD1 were found previously to elicit PDD1 mislocalization, prevented histone H3 dimethylation on K9, abolished removal of internal eliminated sequences (IES), and/or resulted in the production of inviable progeny (*Schwope and Chalker, 2014*). DCL1 (Dicer-like 1) has 5 SCDs, 1 polyQ tract and 8 polyQ/N tracts. DCL1 is required for processing the MIC transcripts to

siRNA-like scan (scn) RNAs, as well as for methylation of histone H3 at Lys 9. This latter modification occurs specifically on sequences (IESs) to be eliminated (*Mochizuki and Gorovsky, 2004*). GIW1 (gentlemen-in-waiting 1) physically directs a mature Argonaute-siRNA complex to the MIC nucleus, thus promoting programmed IES elimination (*Noto et al., 2010*). GIW1 has 9 SCDs and 2 polyQ/N tracts (*Figure 6—source data 33*).

Using IUPred2A (https://iupred2a.elte.hu/plot_new), we found that the Q-rich motifs in most (if not all) of these meiotic proteins are intrinsically disordered. Accordingly, we speculate that, like the C-terminal IDR of mammalian CTTNBP2, the Q-rich motifs in *T. thermophila* meiotic proteins might form tunable proteinaceous condensates to regulate assembly and disassembly of the 'crescents' in ciliate MICs.

## Discussion

We present three unexpected results in this report. First, the Q-rich motifs of several yeast proteins (Rad51-NTD, Rad53-SCD1, Hop1-SCD, Sml1-NTD, Sup35-PND, Ure2-UPD, and New1-NPD) all exhibit autonomous PEE activities. These structurally flexible Q-rich motifs have useful potential for applications in both basic research (e.g. synthetic biology) and biotechnology. Further investigations would prove illuminating as to how these Q-rich motifs exert this PEE function in yeast and whether Q-rich motifs in other eukaryotes also possess similar PEE activities. Second, the reassignment of stop codons to Q in the group I and group II ciliates significantly increases proteome-wide Q usage, leading to massive expansion of structurally flexible or even intrinsically disordered Q-rich motifs. In contrast, reassignments of stop codons to Y, W, or C do not result in higher usages of these three amino acid residues, nor higher percentages of W-, Y-, or C-rich proteins in the three group IV ciliates, respectively. These results are consistent with the notion that, unlike for Q, the Y, W, and C residues are not common in IDRs (*Romero et al., 2001*; *Macossay-Castillo et al., 2019*; *Uversky et al., 2000*). Third, the results in *Table 2* support that a decrease or increase of $CAG^Q$ usage frequency in different eukaryotes is responsible for a reduction or augmentation, respectively, of polyQ instability (or expansion) caused by DNA strand slippage in CAG/CTG trinucleotide repeat regions during DNA replication (*Petruska et al., 1998*; *Mier and Andrade-Navarro, 2021*). Accordingly, it would be interesting to decipher how the molecular mechanism(s) that controls the codon usage bias of $TAA^Q$, $TAG^Q$, $CAA^Q$, and $CAG^Q$ evolved.

Due to their structural flexibility, Q-rich motifs can endow proteins with structural and functional plasticity. Based on previous reports (*Zhou et al., 2019*; *Bondos et al., 2022*) and our findings from this study, Q-rich motifs, such as IDRs, are useful toolkits for generating novel diversity during protein evolution, including by enabling greater protein expression, protein-protein interactions, posttranslational modifications, increased solubility, and tunable stability, among other important traits. This speculation may explain three intriguing phenomena. First, due to higher Q usage, many proteins involved in evolutionarily conserved biological processes in group I and group II ciliates display more diverse amino acid sequences than the respective proteins in other ciliate or non-ciliate species. Accordingly, it is sometimes difficult to identify authentic protein homologs among different ciliates, particularly for group I and group II ciliates. We highlight the example of the 58 proteins involved in xylan catabolysis in *T. thermophila* (*Figure 6—source data 32*). Second, our GO enrichment results reveal that Q-rich motifs prevail in proteins involved in specialized biological processes (*Figure 11* and *Figure 12*). In theory, structurally flexible Q-rich motifs might form various membraneless organelles or proteinaceous condensates via intracellular liquid-liquid phase separation in tunable manners, including protein posttranslational modification, protein-protein interaction, protein-ligand binding, among other processes (*Wright and Dyson, 1999*; *Posey et al., 2018*). A typical example is that the C-terminus of mouse CTTNBP2 facilitates co-condensation of CTTNBP2 with SHANK3 at the postsynaptic density in a $Zn^{2+}$-dependent manner (*Shih et al., 2022*). Third, Borgs are long and linear extrachromosomal DNA sequences in methane-oxidizing *Methanoperedens archaea*, which display the potential to augment methane oxidation (*Al-Shayeb et al., 2022*). A striking feature of Borgs is pervasive tandem direct repeat (TR) regions. TRs in open reading frames (ORFs) are under very strong selective pressure, leading to perfect amino acid TRs (aaTRs) that are commonly IDRs. Notably, aaTRs often contain disorder-promoting amino acids, including Q, N, P, T, E, K, V, D, and S (*Schoelmerich et al., 2023*). Accordingly, distinct evolutionary strategies are employed in different species to alter protein regions that are structurally flexible or even intrinsically disordered. Further investigations are

needed to determine if liquid-liquid phase separation prevails in ciliates with reassigned TAA$^Q$ and TAG$^Q$ codons and/or in the specialized biological processes in various species we have described herein.

## Conclusions

One of the most interesting questions in genome diversity is why many ciliates reassign their nuclear stop codons into amino acids, for example glutamine (Q), tyrosine (Y), tryptophan (W), or cysteine (C). The impacts of such genome-wide alternations had not been well understood. Here, we show that glutamine (Q) is used more frequently in all 10 ciliate species possessing reassigned TAA$^Q$ and TAG$^Q$ codons than in other ciliates and non-ciliate species. The consequence of this preponderance of Q is the massive expansion of proteins harboring structurally flexible or even intrinsically disordered Q-rich motifs. Since Q-rich motifs can endow proteins with structural and functional plasticity and Q-rich-motif proteins are overrepresented in several species-specific or even phylum-specific biological processes, we suggest that Q-rich motifs are useful toolkits for evolutionary novelty.

## Methods

All plasmids, yeast strains, and PCR primers used in this study are listed in *Supplementary file 1a-c*, respectively. Guinea pig antisera against Rad51, and rabbit antisera against phosphorylated Rad51-S$^{12}$Q, phosphorylated Rad51-S$^{30}$Q, and phosphorylated Hop1-T$^{318}$Q were described previously (*Woo et al., 2020*; *Chuang et al., 2012*). The mouse anti-V5 antibody was purchased from BioRad (CA, USA). The rabbit anti-Hsp104 antiserum was kindly provided by Chung Wang (Institute of Molecular Biology, Academia Sinica, Taiwan). Rabbit antisera against phosphorylated Sup35-S$^{17}$Q were raised using the synthetic phosphopeptide N$^{12}$YQQYS$^{(P)}$QNGNQQQGNNR$^{28}$ as an antigen, where S$^{(P)}$ is phosphorylated serine. Phosphopeptide synthesis and animal immunization were conducted by LTK BioLaboratories, Taiwan. Western blotting analyses were performed as described previously (*Woo et al., 2020*; *Chuang et al., 2012*). Quantitative β-galactosidase activity assays were carried out as previously described (*Woo et al., 2020*; *Lin et al., 2010*). The sources of proteome and transcript datasets are described in *Table 1*. All six JavaScript software programs used in this study are listed in *Supplementary file 1d* and publicly available on Github (https://github.com/labASIMBTFWang/AS-Q-rich-motif, copy archived at *Wang, 2024*). The GO enrichment analyses were performed using publicly available data in the GO Resource (http://geneontology.org). The GO identities (ID) of different biological processes, cellular components, and molecular functions, as well as the names of all SCD and polyX proteins in the 26 near-completed eukaryotic proteomes, are listed in *Figure 6—source data 6–31*, respectively. GOfuncR was applied for rigorous statistical testing by conducting standard candidate vs. background enrichment analysis using the hypergeometric test. The raw p-values of over-represented and under-represented GO groups were adjusted according to the Family-wise error rate (FWER).

### Availability of source data files and materials

All experimental materials used in this study are available upon request. The source data analyzed in this study are listed in the supporting information and the source data files. All JavaScript software programs used in this study (*Supplementary file 1*) are available at Github (https://github.com/labASIMBTFWang/AS-Q-rich-motif, *Wang, 2024*).

### Acknowledgements

We thank John O'Brien for English editing, G Titus Brown (0000-0001-6001-2677) for his help in accessing the reassembled transcriptomic dataset in Zendo, Yu-Tang Huang (IMB Computer Room) for maintaining the computer workstation, Josef Loid (Max Perutz Labs, University of Vienna, Austria) for providing the list of 124 proteins involved in *T. thermophila* meiosis, Meng-Chao Yao (IMB, Academia Sinica, Taiwan) for his suggestion to include the MMESTP transcripts of 11 different ciliates in this study

## Additional information

### Funding

| Funder | Grant reference number | Author |
|---|---|---|
| National Science and Technology Council, Taiwan | NSTC 110-2811-B-001-542 | Ting-Fang Wang |
| Academia Sinica | Intramural funding | Ting-Fang Wang Yi-Ping Hsueh |

The funders had no role in study design, data collection and interpretation, or the decision to submit the work for publication.

### Author contributions

Chi-Ning Chuang, Data curation, Formal analysis, Validation, Investigation, Writing – original draft; Hou-Cheng Liu, Data curation, Software, Investigation, Visualization; Tai-Ting Woo, Investigation, Writing – original draft; Ju-Lan Chao, Resources, Investigation; Chiung-Ya Chen, Hisao-Tang Hu, Investigation; Yi-Ping Hsueh, Funding acquisition, Writing – original draft; Ting-Fang Wang, Conceptualization, Resources, Data curation, Software, Formal analysis, Supervision, Funding acquisition, Validation, Investigation, Visualization, Methodology, Writing – original draft, Project administration, Writing - review and editing

### Author ORCIDs

Tai-Ting Woo ⓘ https://orcid.org/0000-0002-9717-1142
Yi-Ping Hsueh ⓘ http://orcid.org/0000-0002-0866-6275
Ting-Fang Wang ⓘ http://orcid.org/0000-0001-6306-9505

Reviewer #2 (Public Review): https://doi.org/10.7554/eLife.91405.3.sa1
Author Response https://doi.org/10.7554/eLife.91405.3.sa2

---

## Additional files

### Supplementary files

- MDAR checklist
- Supplementary file 1. Plasids, yeast strains, PCR primers, and home-made software tools.
- Source data 1. Raw data and statistical data for *Figures 1–5*.

### Data availability

All data generated and analysed during this study are included in the manuscript, supporting tables, and source data files. All softwares used in this study are publicly available at Github: https://github.com/labASIMBTFWang/AS-Q-rich-motif (copy archived at *Wang, 2024*). Software has been licensed with an open source license.

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
