## [Editor Report · eLife assessment]

This study presents **useful** results on glutamine-rich motifs in relation to protein expression and alternative genetic codes. The **solid** data are based on bioinformatic approaches that are employed to systematically uncover sequence features associated with proteome-wide amino acid distribution and biological processes.

---

## [Referee Report · Reviewer #2 (Public Review)]

Summary:

This study seeks to understand the connection between protein sequence and function in disordered regions enriched in polar amino acids (specifically Q, N, S and T). While the authors suggest that specific motifs facilitate protein-enhancing activities, their findings are correlative, and the evidence is incomplete. Similarly, the authors propose that the re-assignment of stop codons to glutamine-encoding codons underlies the greater user of glutamine in a subset of ciliates, but again, the conclusions here are, at best, correlative. The authors perform extensive bioinformatic analysis, with detailed (albeit somewhat ad hoc) discussion on a number of proteins. Overall, the results presented here are interesting but are unable to exclude competing hypotheses.

Strengths:

Following up on previous work, the authors wish to uncover a mechanism associated with poly-Q and SCD motifs explaining proposed protein expression-enhancing activities. They note that these motifs often occur IDRs and hypothesize that structural plasticity could be capitalized upon as a mechanism of diversification in evolution. To investigate this further, they employ bioinformatics to investigate the sequence features of proteomes of 27 eukaryotes. They deepen their sequence space exploration uncovering sub-phylum-specific features associated with species in which a stop-codon substitution has occurred. The authors propose this stop-codon substitution underlies an expansion of ploy-Q repeats and increased glutamine distribution.

Weaknesses:

The authors were provided with a series of suggested changes to improve clarity, and a series of concerns raised. Some of these have been addressed but many have not. At this point, I do not see my role as telling the authors how to re-write their manuscript, but many of the concerns raised in my original review remain, and the authors have done little to allay those concerns in their revisions.

---

## [Author Response]

The following is the authors’ response to the original reviews.

**eLife assessment**
This study presents potentially valuable results on glutamine-rich motifs in relation to protein expression and alternative genetic codes. The author's interpretation of the results is so far only supported by incomplete evidence, due to a lack of acknowledgment of alternative explanations, missing controls and statistical analysis and writing unclear to non experts in the field. These shortcomings could be at least partially overcome by additional experiments, thorough rewriting, or both.

We thank both the Reviewing Editor and Senior Editor for handling this manuscript.

Based on your suggestions, we have provided controls, performed statistical analysis, and rewrote our manuscript. The revised manuscript is significantly improved and more accessible to non-experts in the field.

**Reviewer #1 (Public Review):**
SummaryThis work contains 3 sections. The first section describes how protein domains with SQ motifs can increase the abundance of a lacZ reporter in yeast. The authors call this phenomenon autonomous protein expression-enhancing activity, and this finding is well supported. The authors show evidence that this increase in protein abundance and enzymatic activity is not due to changes in plasmid copy number or mRNA abundance, and that this phenomenon is not affected by mutants in translational quality control. It was not completely clear whether the increased protein abundance is due to increased translation or to increased protein stability.In section 2, the authors performed mutagenesis of three N-terminal domains to study how protein sequence changes protein stability and enzymatic activity of the fusions. These data are very interesting, but this section needs more interpretation. It is not clear if the effect is due to the number of S/T/Q/N amino acids or due to the number of phosphorylation sites.In section 3, the authors undertake an extensive computational analysis of amino acid runs in 27 species. Many aspects of this section are fascinating to an expert reader. They identify regions with poly-X tracks. These data were not normalized correctly: I think that a null expectation for how often poly-X track occur should be built for each species based on the underlying prevalence of amino acids in that species. As a result, I believe that the claim is not well supported by the data.StrengthsThis work is about an interesting topic and contains stimulating bioinformatics analysis. The first two sections, where the authors investigate how S/T/Q/N abundance modulates protein expression level, is well supported by the data. The bioinformatics analysis of Q abundance in ciliate proteomes is fascinating. There are some ciliates that have repurposed stop codons to code for Q. The authors find that in these proteomes, Q-runs are greatly expanded. They offer interesting speculations on how this expansion might impact protein function.WeaknessAt this time, the manuscript is disorganized and difficult to read. An expert in the field, who will not be distracted by the disorganization, will find some very interesting results included. In particular, the order of the introduction does not match the rest of the paper.In the first and second sections, where the authors investigate how S/T/Q/N abundance modulates protein expression levels, it is unclear if the effect is due to the number of phosphorylation sites or the number of S/T/Q/N residues.

There are three reasons why the number of phosphorylation sites in the Q-rich motifs is not relevant to their autonomous protein expression-enhancing (PEE) activities:

First, we have reported previously that phosphorylation-defective Rad51-NTD (Rad51-3SA) and wild-type Rad51-NTD exhibit similar autonomous PEE activity. Mec1/Tel1-dependent phosphorylation of Rad51-NTD antagonizes the proteasomal degradation pathway, increasing the half-life of Rad51 from ∼30 min to ≥180 min (1). (page 1, lines 11-14)

Second, in our preprint manuscript, we have already shown that phosphorylation-defective Rad53-SCD1 (Rad51-SCD1-5STA) also exhibits autonomous PEE activity similar to that of wild-type Rad53-SCD (Figure 2D, Figure 4A and Figure 4C). We have highlighted this point in our revised manuscript (page 9, lines 19-21).

Third, as revealed by the results of Figure 4, it is the percentages, and not the numbers, of S/T/Q/N residues that are correlated with the PEE activities of Q-rich motifs.

The authors also do not discuss if the N-end rule for protein stability applies to the lacZ reporter or the fusion proteins.

The autonomous PEE function of S/T/Q-rich NTDs is unlikely to be relevant to the N-end rule. The N-end rule links the in vivo half-life of a protein to the identity of its N-terminal residues. In *S. cerevisiae*, the N-end rule operates as part of the ubiquitin system and comprises two pathways. First, the Arg/N-end rule pathway, involving a single N-terminal amidohydrolase Nta1, mediates deamidation of N-terminal asparagine (N) and glutamine (Q) into aspartate (D) and glutamate (E), which in turn are arginylated by a single Ate1 R-transferase, generating the Arg/N degron. N-terminal R and other primary degrons are recognized by a single N-recognin Ubr1 in concert with ubiquitin-conjugating Ubc2/Rad6. Ubr1 can also recognize several other N-terminal residues, including lysine (K), histidine (H), phenylalanine (F), tryptophan (W), leucine (L) and isoleucine (I) (68-70). Second, the Ac/N-end rule pathway targets proteins containing N-terminally acetylated (Ac) residues. Prior to acetylation, the first amino acid methionine (M) is catalytically removed by Met-aminopeptidases (MetAPs), unless a residue at position 2 is non-permissive (too large) for MetAPs. If a retained N-terminal M or otherwise a valine (V), cysteine (C), alanine (A), serine (S) or threonine (T) residue is followed by residues that allow N-terminal acetylation, the proteins containing these AcN degrons are targeted for ubiquitylation and proteasome-mediated degradation by the Doa10 E3 ligase (71).

The PEE activities of these S/T/Q-rich domains are unlikely to arise from counteracting the N-end rule for two reasons. First, the first two amino acid residues of Rad51-NTD, Hop1-SCD, Rad53-SCD1, Sup35-PND, Rad51-ΔN, and LacZ-NVH are MS, ME, ME, MS, ME, and MI, respectively, where M is methionine, S is serine, E is glutamic acid and I is isoleucine. Second, Sml1-NTD behaves similarly to these N-terminal fusion tags, despite its methionine and glutamine (MQ) amino acid signature at the N-terminus. (Page 12, line 3 to page 13, line 2)

The most interesting part of the paper is an exploration of S/T/Q/N-rich regions and other repetitive AA runs in 27 proteomes, particularly ciliates. However, this analysis is missing a critical control that makes it nearly impossible to evaluate the importance of the findings. The authors find the abundance of different amino acid runs in various proteomes. They also report the background abundance of each amino acid. They do not use this background abundance to normalize the runs of amino acids to create a null expectation from each proteome. For example, it has been clear for some time (Ruff, 2017; Ruff et al., 2016) that *Drosophila* contains a very high background of Q's in the proteome and it is necessary to control for this background abundance when finding runs of Q's.

We apologize for not explaining sufficiently well the topic eliciting this reviewer’s concern in our preprint manuscript. In the second paragraph of page 14, we cite six references to highlight that SCDs are overrepresented in yeast and human proteins involved in several biological processes (5, 43) and that polyX prevalence differs among species (79-82).

We will cite a reference by Kiersten M. Ruff in our revised manuscript (38).

K. M. Ruff, J. B. Warner, A. Posey and P. S. Tan (2017) Polyglutamine length dependent structural properties and phase behavior of huntingtin exon1. Biophysical Journal 112, 511a.

The authors could easily address this problem with the data and analysis they have already collected. However, at this time, without this normalization, I am hesitant to trust the lists of proteins with long runs of amino acid and the ensuing GO enrichment analysis.Ruff KM. 2017. Washington University in St.Ruff KM, Holehouse AS, Richardson MGO, Pappu RV. 2016. Proteomic and Biophysical Analysis of Polar Tracts. Biophys J 110:556a.

We thank Reviewer #1 for this helpful suggestion and now address this issue by means of a different approach described below.

Based on a previous study (43), we applied seven different thresholds to seek both short and long, as well as pure and impure, polyX strings in 20 different representative near-complete proteomes, including 4X (4/4), 5X (4/5-5/5), 6X (4/6-6/6), 7X (4/7-7/7), 8-10X (≥50%X), 11-10X (≥50%X) and ≥21X (≥50%X).

To normalize the runs of amino acids and create a null expectation from each proteome, we determined the ratios of the overall number of X residues for each of the seven polyX motifs relative to those in the entire proteome of each species, respectively. The results of four different polyX motifs are shown in our revised manuscript, i.e., polyQ (Figure 7), polyN (Figure 8), polyS (Figure 9) and polyT (Figure 10). Thus, polyX prevalence differs among species and the overall X contents of polyX motifs often but not always correlate with the X usage frequency in entire proteomes (43).

Most importantly, our results reveal that, compared to Stentor coeruleus or several non-ciliate eukaryotic organisms (e.g., *Plasmodium falciparum*, *Caenorhabditis elegans*, *Danio rerio*, *Mus musculus* and *Homo sapiens*), the five ciliates with reassigned TAAQ and TAGQ codons not only have higher Q usage frequencies, but also more polyQ motifs in their proteomes (Figure 7). In contrast, polyQ motifs prevail in Candida albicans, Candida tropicalis, *Dictyostelium* discoideum, Chlamydomonas reinhardtii, *Drosophila melanogaster* and Aedes aegypti, though the Q usage frequencies in their entire proteomes are not significantly higher than those of other eukaryotes (Figure 1). Due to their higher N usage frequencies, *Dictyostelium* discoideum, *Plasmodium falciparum* and Pseudocohnilembus persalinus have more polyN motifs than the other 23 eukaryotes we examined here (Figure 8). Generally speaking, all 26 eukaryotes we assessed have similar S usage frequencies and percentages of S contents in polyS motifs (Figure 9). Among these 26 eukaryotes, *Dictyostelium* discoideum possesses many more polyT motifs, though its T usage frequency is similar to that of the other 25 eukaryotes (Figure 10).

In conclusion, these new normalized results confirm that the reassignment of stop codons to Q indeed results in both higher Q usage frequencies and more polyQ motifs in ciliates.

**Reviewer #2 (Public Review):**
Summary:This study seeks to understand the connection between protein sequence and function in disordered regions enriched in polar amino acids (specifically Q, N, S and T). While the authors suggest that specific motifs facilitate protein-enhancing activities, their findings are correlative, and the evidence is incomplete. Similarly, the authors propose that the re-assignment of stop codons to glutamine-encoding codons underlies the greater user of glutamine in a subset of ciliates, but again, the conclusions here are, at best, correlative. The authors perform extensive bioinformatic analysis, with detailed (albeit somewhat ad hoc) discussion on a number of proteins. Overall, the results presented here are interesting, but are unable to exclude competing hypotheses.Strengths:Following up on previous work, the authors wish to uncover a mechanism associated with poly-Q and SCD motifs explaining proposed protein expression-enhancing activities. They note that these motifs often occur IDRs and hypothesize that structural plasticity could be capitalized upon as a mechanism of diversification in evolution. To investigate this further, they employ bioinformatics to investigate the sequence features of proteomes of 27 eukaryotes. They deepen their sequence space exploration uncovering sub-phylum-specific features associated with species in which a stop-codon substitution has occurred. The authors propose this stop-codon substitution underlies an expansion of ploy-Q repeats and increased glutamine distribution.Weaknesses:The preprint provides extensive, detailed, and entirely unnecessary background information throughout, hampering reading and making it difficult to understand the ideas being proposed.

The introduction provides a large amount of detailed background that appears entirely irrelevant for the paper. Many places detailed discussions on specific proteins that are likely of interest to the authors occur, yet without context, this does not enhance the paper for the reader.

The paper uses many unnecessary, new, or redefined acronyms which makes reading difficult. As examples:1. Prion forming domains (PFDs). Do the authors mean prion-like domains (PLDs), an established term with an empirical definition from the PLAAC algorithm? If yes, they should say this. If not, they must define what a prion-forming domain is formally.

The N-terminal domain (1-123 amino acids) of *S. cerevisiae* Sup35 was already referred to as a “prion forming domain (PFD)” in 2006 (48). Since then, PFD has also been employed as an acronym in other yeast prion papers (Cox, B.S. et al. 2007; Toombs, T. et al. 2011).

B. S. Cox, L. Byrne, M. F., Tuite, Protein Stability. Prion 1, 170-178 (2007).J. A. Toombs, N. M. Liss, K. R. Cobble, Z. Ben-Musa, E. D. Ross, [PSI+] maintenance is dependent on the composition, not primary sequence, of the oligopeptide repeat domain. PLoS One 6, e21953 (2011).

1. SCD is already an acronym in the IDP field (meaning sequence charge decoration) - the authors should avoid this as their chosen acronym for Serine(S) / threonine (T)-glutamine (Q) cluster domains. Moreover, do we really need another acronym here (we do not).

SCD was first used in 2005 as an acronym for the Serine (S)/threonine (T)-glutamine (Q) cluster domain in the DNA damage checkpoint field (4). Almost a decade later, SCD became an acronym for “sequence charge decoration” (Sawle, L. et al. 2015; Firman, T. et al. 2018).

L. Sawle and K, Ghosh, A theoretical method to compute sequence dependent configurational properties in charged polymers and proteins. J. Chem Phys. 143, 085101(2015).

T. Firman and Ghosh, K. Sequence charge decoration dictates coil-globule transition in intrinsically disordered proteins. J. Chem Phys. 148, 123305 (2018).

1. Protein expression-enhancing (PEE) - just say expression-enhancing, there is no need for an acronym here.

Thank you. Since we have shown that the addition of Q-rich motifs to LacZ affects protein expression rather than transcription, we think it is better to use the “PEE” acronym.

The results suggest autonomous protein expression-enhancing activities of regions of multiple proteins containing Q-rich and SCD motifs. Their definition of expression-enhancing activities is vague and the evidence they provide to support the claim is weak. While their previous work may support their claim with more evidence, it should be explained in more detail. The assay they choose is a fusion reporter measuring beta-galactosidase activity and tracking expression levels. Given the presented data they have shown that they can drive the expression of their reporters and that beta gal remains active, in addition to the increase in expression of fusion reporter during the stress response. They have not detailed what their control and mock treatment is, which makes complete understanding of their experimental approach difficult. Furthermore, their nuclear localization signal on the tag could be influencing the degradation kinetics or sequestering the reporter, leading to its accumulation and the appearance of enhanced expression. Their evidence refuting ubiquitin-mediated degradation does not have a convincing control.

Although this reviewer’s concern regarding our use of a nuclear localization signal on the tag is understandable, we are confident that this signal does not bias our findings for two reasons. First, the negative control LacZ-NV also possesses the same nuclear localization signal (Figure 1A, lane 2). Second, another fusion target, Rad51-ΔN, does not harbor the NVH tag (Figure 1D, lanes 3-4). Compared to wild-type Rad51, Rad51-ΔN is highly labile. In our previous study, removal of the NTD from Rad51 reduced by ~97% the protein levels of corresponding Rad51-ΔN proteins relative to wild-type (1).

Based on the experimental results, the authors then go on to perform bioinformatic analysis of SCD proteins and polyX proteins. Unfortunately, there is no clear hypothesis for what is being tested; there is a vague sense of investigating polyX/SCD regions, but I did not find the connection between the first and section compelling (especially given polar-rich regions have been shown to engage in many different functions). As such, this bioinformatic analysis largely presents as many lists of percentages without any meaningful interpretation. The bioinformatics analysis lacks any kind of rigorous statistical tests, making it difficult to evaluate the conclusions drawn. The methods section is severely lacking. Specifically, many of the methods require the reader to read many other papers. While referencing prior work is of course, important, the authors should ensure the methods in this paper provide the details needed to allow a reader to evaluate the work being presented. As it stands, this is not the case.

Thank you. As described in detail below, we have now performed rigorous statistical testing using the GofuncR package (Figure 11, Figure 12 and DS7-DS32).

Overall, my major concern with this work is that the authors make two central claims in this paper (as per the Discussion). The authors claim that Q-rich motifs enhance protein expression. The implication here is that Q-rich motif IDRs are special, but this is not tested. As such, they cannot exclude the competing hypothesis ("N-terminal disordered regions enhance expression").

In fact, “N-terminal disordered regions enhance expression” exactly summarizes our hypothesis.

On pages 12-13 and Figure 4 of our preprint manuscript, we explained our hypothesis in the paragraph entitled “The relationship between PEE function, amino acid contents, and structural flexibility”.

The authors also do not explore the possibility that this effect is in part/entirely driven by mRNA-level effects (see Verma Na Comms 2019).

As pointed out by the first reviewer, we present evidence that the increase in protein abundance and enzymatic activity is not due to changes in plasmid copy number or mRNA abundance (Figure 2), and that this phenomenon is not affected in translational quality control mutants (Figure 3).

As such, while these observations are interesting, they feel preliminary and, in my opinion, cannot be used to draw hard conclusions on how N-terminal IDR sequence features influence protein expression. This does not mean the authors are necessarily wrong, but from the data presented here, I do not believe strong conclusions can be drawn. That re-assignment of stop codons to Q increases proteome-wide Q usage. I was unable to understand what result led the authors to this conclusion.My reading of the results is that a subset of ciliates has re-assigned UAA and UAG from the stop codon to Q. Those ciliates have more polyQ-containing proteins. However, they also have more polyN-containing proteins and proteins enriched in S/T-Q clusters. Surely if this were a stop-codon-dependent effect, we'd ONLY see an enhancement in Q-richness, not a corresponding enhancement in all polar-rich IDR frequencies? It seems the better working hypothesis is that free-floating climate proteomes are enriched in polar amino acids compared to sessile ciliates.

We thank this reviewer for raising this point, however her/his comments are not supported by the results in Figure 7.

Regardless, the absence of any kind of statistical analysis makes it hard to draw strong conclusions here.

We apologize for not explaining more clearly the results of Tables 5-7 in our preprint manuscript.

To address the concerns about our GO enrichment analysis by both reviewers, we have now performed rigorous statistical testing for SCD and polyQ protein overrepresentation using the GOfuncR package (https://bioconductor.org/packages/release/bioc/html/GOfuncR.html). GOfuncR is an R package program that conducts standard candidate vs. background enrichment analysis by means of the hypergeometric test. We then adjusted the raw p-values according to the Family-wise error rate (FWER). The same method had been applied to GO enrichment analysis of human genomes (89).

The results presented in Figure 11 and Figure 12 (DS7-DS32) support our hypothesis that Q-rich motifs prevail in proteins involved in specialized biological processes, including *Saccharomyces cerevisiae* RNA-mediated transposition, Candida albicans filamentous growth, peptidyl-glutamic acid modification in ciliates with reassigned stop codons (TAAQ and TAGQ), *Tetrahymena thermophila* xylan catabolism, *Dictyostelium* discoideum sexual reproduction, *Plasmodium falciparum* infection, as well as the nervous systems of *Drosophila melanogaster*, *Mus musculus*, and *Homo sapiens* (78). In contrast, peptidyl-glutamic acid modification and microtubule-based movement are not overrepresented with Q-rich proteins in Stentor coeruleus, a ciliate with standard stop codons.

**Recommendations for the authors:**
Please note that you control which revisions to undertake from the public reviews and recommendations for the authors.
**Reviewer #1 (Recommendations For The Authors):**
The order of paragraphs in the introduction was very difficult to follow. Each paragraph was clear and easy to understand, but the order of paragraphs did not make sense to this reader. The order of events in the abstract matches the order of events in the results section. However, the order of paragraphs in the introduction is completely different and this was very confusing. This disordered list of facts might make sense to an expert reader but makes it hard for a non-expert reader to understand.

Apologies. We endeavored to improve the flow of our revised manuscript to make it more readable.

The section beginning on pg 12 focused on figures 4 and 5 was very interesting and highly promising. However, it was initially hard for me to tell from the main text what the experiment was. Please add to the text an explanation of the experiment, because it is hard to figure out what was going on from the figures alone. Figure 4 is fantastic, but would be improved by adding error bars and scaling the x-axis to be the same in panels B,C,D.

Thank you for this recommendation. We have now scaled both the x-axis and y-axis equivalently in panels B, C and D of Figure 4. Error bars are too small to be included.

It is hard to tell if the key variable is the number of S/T/Q/N residues or the number of phosphosites. I think a good control would be to add a regression against the number of putative phosphosites. The sequences are well designed. I loved this part but as a reader, I need more interpretation about why it matters and how it explains the PEE.

As described above, we have shown that the number of phosphorylation sites in the Q-rich motifs is not relevant to their autonomous protein expression-enhancing (PEE) activities.

I believe that the prevalence of polyX runs is not meaningful without normalizing for the background abundance of each amino acid. The proteome-wide abundance and the assumption that amino acids occur independently can be used to form a baseline expectation for which runs are longer than expected by chance. I think Figures 6 and 7 should go into the supplement and be replaced in the main text with a figure where Figure 6 is normalized by Figure 7. For example in *P. falciparum*, there are many N-runs (Figure 6), but the proteome has the highest fraction of N’s (Figure 7).

Thank you for these suggestions. The three figures in our preprint manuscript (Figures 6-8) have been moved into the supplementary information (Figures S1-S3). For normalization, we have provided four new figures (Figures 7-10) in our revised manuscript.

The analysis of ciliate proteomes was fascinating. I am particularly interested in the GO enrichment for “peptidyl-glutamic acid modification” (pg 20) because these enzymes might be modifying some of Q’s in the Q-runs. I might be wrong about this idea or confused about the chemistry. Do these ciliates live in Q-rich environments? Or nitrogen rich environments?

Polymeric modifications (polymodifications) are a hallmark of C-terminal tubulin tails, whereas secondary peptide chains of glutamic acids (polyglutamylation) and glycines (polyglycylation) are catalyzed from the γ-carboxyl group of primary chain glutamic acids. It is not clear if these enzymes can modify some of the Q’s in the Q-runs.

To our knowledge, ciliates are abundant in almost every liquid water environment, i.e., oceans/seas, marine sediments, lakes, ponds, and rivers, and even soils.

I think you should include more discussion about how the codons that code for Q’s are prone to slippage during DNA replication, and thus many Q-runs are unstable and expand (e.g. Huntington’s Disease). The end of pg 24 or pg 25 would be good places.

We thank the reviewer for these comments.

PolyQ motifs have a particular length-dependent codon usage that relates to strand slippage in CAG/CTG trinucleotide repeat regions during DNA replication. In most organisms having standard genetic codons, Q is encoded by CAGQ and CAAQ. Here, we have determined and compared proteome-wide Q contents, as well as the CAGQ usage frequencies (i.e., the ratio between CAGQ and the sum of CAGQ, CAGQ, TAAQ, and TAGQ).

Our results reveal that the likelihood of forming long CAG/CTG trinucleotide repeats are higher in five eukaryotes due to their higher CAGQ usage frequencies, including *Drosophila melanogaster* (86.6% Q), *Danio rerio* (74.0% Q), *Mus musculus* (74.0% Q), *Homo sapiens* (73.5% Q), and Chlamydomonas reinhardtii (87.3% Q) (orange background, Table 2). In contrast, another five eukaryotes that possess high numbers of polyQ motifs (i.e., *Dictyostelium* discoideum, Candida albicans, Candida tropicalis, *Plasmodium falciparum* and Stentor coeruleus) (Figure 1) utilize more CAAQ (96.2%, 84.6%, 84.5%, 86.7% and 75.7%) than CAAQ (3.8%, 15.4%, 15.5%, 13.3% and 24.3%), respectively, to avoid the formation of long CAG/CTG trinucleotide repeats (green background, Table 2). Similarly, all five ciliates with reassigned stop codons (TAAQ and TAGQ) have low CAGQ usage frequencies (i.e., from 3.8% Q in Pseudocohnilembus persalinus to 12.6% Q in Oxytricha trifallax) (red font, Table 2). Accordingly, the CAG-slippage mechanism might operate more frequently in Chlamydomonas reinhardtii, *Drosophila melanogaster*, *Danio rerio*, *Mus musculus* and *Homo sapiens* than in *Dictyostelium* discoideum, Candida albicans, Candida tropicalis, *Plasmodium falciparum*, Stentor coeruleus and the five ciliates with reassigned stop codons (TAAQ and TAGQ).

**Author response table 1. sa2table1:** Usage frequencies of TAA*, TAG*, TAAQ, TAGQ, CAAQ and CAGQ codonsin the entire proteomes of 20 different organisms.

Species_name	CAA	CAG	TAA	TAG
*Saccharomyces cerevisiae* S288c	2.73(62.6%Q)	1.21(37.4%Q)	0.11	0.05
Candida albicans	3.57(84.6%Q)	0.65(15.4%Q)	0.1	0.05
Candida auris	1.81(46.1%Q)	2.12(53.9%Q)	0.08	0.06
Candida tropicalis	3.61(84.5%Q)	0.66(15.5%Q)	0.1	0.07
*Neurospora crassa*	1.70(39.5%Q)	2.60(60.5%Q)	0.06	0.05
Magnaporthe oryzae	1.37(33.7%Q)	2.69(66.3%Q)	0.06	0.07
Trichoderma reesei	1.17(28.4%Q)	2.95(71.6%Q)	0.06	0.06
Cryptococcus neoformans	2.06(53.5%Q)	1.79(46.5%Q)	0.07	0.06
*Ustilago maydis*	1.82(41.3%Q)	2.61(58.7%Q)	0.04	0.05
Taiwanofungus camphoratus	1.57(41.8%Q)	2.19(58.2%Q)	0.05	0.06
*Dictyostelium* discoideum	4.86(96.2%Q)	0.19(3.8%Q)	0.16	0.01
*Plasmodium falciparum*	2.42(86.7%Q)	0.37(13.3%Q)	0.09	0.01
*Drosophila melanogaster*	1.56(13.4%Q)	3.61(86.6%Q)	0.08	0.07
Aedes aegypti	1.76(40.6%Q)	2.58(59.4%Q)	0.11	0.07
*Caenorhabditis elegans*	2.74(65.6%Q)	1.44(34.4%Q)	0.16	0.06
*Danio rerio*	1.18(26.0%Q)	3.35(74.0%Q)	0.11	0.06
*Mus musculus*	1.20(26.0%Q)	3.41(74.0%Q)	0.1	0.08
*Homo sapiens*	1.23(26.5%Q)	3.42(73.5%Q)	0.1	0.08
*Arabidopsis thaliana*	1.94(56.1%Q)	1.52(43.9%Q)	0.09	0.05
*Chlamydomonas reinhardtii*	0.59(12.7%Q)	4.05(87.3%Q)	0.03	0.04
*Tetrahymena thermophila*	2.04(21.2%Q)	0.48(5.0%Q)	5.46(56.8%Q)	1.63(17.0%Q)
Paramecium tetraurelia	2.54(27.9%Q)	0.57(6.3%Q)	4.53(46.7%Q)	1.48(16.2%Q)
Oxytricha trifallax	2.68(29.9%Q)	1.07(12.0%Q)	3.63(40.6%Q)	1.57(17.5%Q)
Stylonychia lemnae	2.26(21.1%Q)	1.05(12.6%Q)	3.22(38.6%Q)	1.81(21.7%Q)
Pseudocohnilembus persalinus	1.76(18.0%Q)	0.37(3.8%Q)	7.36(76.0%Q)	1.39(14.4%Q)
Stentor coeruleus	2.77(75.7%Q)	0.89(24.3%Q)	0.16	0.08

Pg 7, paragraph 2 has no direction. Please add the conclusion of the paragraph to the first sentence.

This paragraph has been moved to the “Introduction” section” of the revised manuscript.

Pg 8, I suggest only mentioning the PFDs used in the experiments. The rest are distracting.

We have addressed this concern above.

Pg 12. Please revise the "The relationship...." text to explain the experiment.

We apologize for not explaining this topic sufficiently well in our preprint manuscript.

SCDs are often structurally flexible sequences (4) or even IDRs. Using IUPred2A (https://iupred2a.elte.hu/plot_new), a web-server for identifying disordered protein regions (88), we found that Rad51-NTD (1-66 a.a.) (1), Rad53-SCD1 (1-29 a.a.) and Sup35-NPD (1-39 a.a.) are highly structurally flexible. Since a high content of serine (S), threonine (T), glutamine (Q), asparanine (N) is a common feature of IDRs (17-20), we applied alanine scanning mutagenesis approach to reduce the percentages of S, T, Q or N in Rad51-NTD, Rad53-SCD1 or Sup35-NPD, respectively. As shown in Figure 4 and Figure 5, there is a very strong positive relationship between STQ and STQN amino acid percentages and β-galactosidase activities. (Page 13, lines 5-10)

Pg 13, first full paragraph, "Futionally, IDRs..." I think this paragraph belongs in the Discussion.

This paragraph is now in the “Introduction” section (Page 5, Lines 11-15).

Pg. 15, I think the order of paragraphs should be swapped.

These paragraphs have been removed or rewritten in the “Introduction section” of our revised manuscript.

Pg 17 (and other parts) I found the lists of numbers and percentages hard to read and I think you should refer readers to the tables.

Thank you. In the revised manuscript, we have avoided using lists of numbers and percentages, unless we feel they are absolutely essential.

Pg. 19 please add more interpretation to the last paragraph. It is very cool but I need help understanding the result. Are these proteins diverging rapidly? Perhaps this is a place to include the idea of codon slippage during DNA replication.

Thank you. The new results in Table 2 indicate that the CAG-slippage mechanism is unlikely to operate in ciliates with reassigned stop codons (TAAQ and TAGQ).

Pg 24. "Based on our findings from this study, we suggest that Q-rich motifs are useful toolkits for generating novel diversity during protein evolution, including by enabling greater protein expression, protein-protein interactions, posttranslational modifications, increased solubility, and tunable stability, among other important traits." This idea needs to be cited. Keith Dunker has written extensively about this idea as have others. Perhaps also discuss why Poly Q rich regions are different from other IDRs and different from other IDRs that phase-separate.

Agreed, we have cited two of Keith Dunker’s papers in our revised manuscript (73, 74).

Minor notes:Please define Borg genomes (pg 25).

Borgs are long extrachromosomal DNA sequences in methane-oxidizing Methanoperedens archaea, which display the potential to augment methane oxidation (101). They are now described in our revised manuscript. (Page 15, lines 12-14)

**Reviewer #2 (Recommendations For The Authors):**
The authors dance around disorder but never really quantify or show data. This seems like a strange blindspot.

We apologize for not explaining this topic sufficiently well in our preprint manuscript. We have endeavored to do so in our revised manuscript.

The authors claim the expression enhancement is "autonomous," but they have not ruled things out that would make it not autonomous.

Evidence of the “autonomous” nature of expression enhancement is presented in Figure 1, Figure 4, and Figure 5 of the preprint manuscript.

Recommendations for improving the writing and presentation.The title does not recapitulate the entire body of work. The first 5 figures are not represented by the title in any way, and indeed, I have serious misgivings as to whether the conclusion stated in the title is supported by the work. I would strongly suggest the authors change the title.Figure 2 could be supplemental.

Thank you. We think it is important to keep Figure 2 in the text.

Figures 4 and 5 are not discussed much or particularly well.

This reviewer’s opinion of Figure 4 and Figure 5 is in stark contrast to those of the first reviewer.

The introduction, while very thorough, takes away from the main findings of the paper. It is more suited to a review and not a tailored set of minimal information necessary to set up the question and findings of the paper. The question that the authors are after is also not very clear.

Thank you. The entire “Introduction” section has been extensively rewritten in the revised manuscript.

Schematics of their fusion constructs and changes to the sequence would be nice, even if supplemental.

Schematics of the fusion constructs are provided in Figure 1A.

The methods section should be substantially expanded.

The method section in the revised manuscript has been rewritten and expanded. The six Javascript programs used in this work are listed in Table S4.

The text is not always suited to the general audience and readership of eLife.

We have now rewritten parts of our manuscript to make it more accessible to the broad readership of eLife.

In some cases, section headers really don't match what is presented, or there is no evidence to back the claim.

The section headers in the revised manuscript have been corrected.

A lot of the listed results in the back half of the paper could be a supplemental table, listing %s in a paragraph (several of them in a row) is never nice

Acknowledged. In the revised manuscript, we have removed almost all sentences listing %s.

Minor corrections to the text and figures.There is a reference to table 1 multiple times, and it seems that there is a missing table. The current table 1 does not seem to be the same table referred to in some places throughout the text.

Apologies for this mistake, which we have now corrected in our revised manuscript.

In some places its not clear where new work is and where previous work is mentioned. It would help if the authors clearly stated "In previous work...."

Acknowledged. We have corrected this oversight in our revised manuscript.

Not all strains are listed in the strain table (KO's in figure 3 are not included)

Apologies, we have now corrected Table S2, as suggested by this reviewer.

**Author response table 2. sa2table2:** *S. cerevisiae* strains used in this study.

Name	Genotype
WHY13008	MATa, ho, leu2, ura3, his 4-X::LEU2-(NgoMIV;+ ori)-URA3, ERGI(SpeI),RAD51::hphMX4
WHY13283	MATa, ho, leu2, ura3, his 4-X::LEU2-(NgoMIV;+ ori)-URA3, ERGI(SpeI),rad51A::hphMX4
WHY13416	MATa, ho, leu2, ura3, his4-X::LEU2-(NgoMIV;+ori)-URA3, ERGI(SpeI),rad51 /_\N::hphMX4
WHY13744	
WHY13743	
WHY13741	MATa, ho::LYS2, leu2, ura3, lys2, HIS4::LEU2-(BamHI;+ori), ERGI(SalI),SUP35 ^(PND ")-rad51AN::hphMX4"
WHY10271	MATa, ho::hisG, lys2, leu2::hisG, arg4-nsp, ura3
WHY13970	MATa his 3Delta1, leu 2Delta0, met15 Delta0, ura3 Delta0
WHY13785	MATa his 3Delta1, leu 2Delta0, met 15 Delta0, ura 3Delta0, hsp104::kanMX4
WHY14126	MATa his 3Delta1, leu 2Delta0, met 15 Delta0, ura 3Delta0, new 1::kan MX4
WHY14129	MATa his 3Delta1, leu 2Delta0, met 15 Delta0, ura 3Delta0, doa 4::kan MX4
WHY14227	
WHY13989	MATa his 3Delta1, leu 2Delta0, met 15 Delta0, ura 3Delta0,san 1:'kan MX4
WHY14132	MATa his 3Delta1, leu 2Delta0, met 15 Delta0, ura 3Delta0, oaz 1:'kan MX4